# Hierarchical Object-Oriented POMDP Planning for Object Rearrangement

## Abstract

We present an online planning approach and a new benchmark dataset for solving multi-object rearrangement problems in partially observable, multi-room environments. Current object rearrangement solutions, primarily based on Reinforcement Learning or hand-coded planning methods, often lack adaptability to diverse challenges. To address this limitation, we propose a Hierarchical Object-Oriented Partially Observed Markov Decision Process (HOO-POMDP) planner that leverages object-factored belief representations for efficient multi-object rearrangement. This approach comprises of (a) an object-oriented POMDP planner generating sub-goals, (b) a set of low-level policies for sub-goal achievement, and (c) an abstraction system converting the continuous low-level world into a representation suitable for abstract planning. To enable rigorous evaluation of rearrangement challenges, we introduce MultiRoomR, a comprehensive benchmark featuring diverse multi-room environments with varying degrees of partial observability (10-30% initial visibility), blocked paths, obstructed goals, and multiple objects (10-20) distributed across 2-4 rooms. Experiments demonstrate that our system effectively handles these complex scenarios while maintaining robust performance even with imperfect perception, achieving promising results across both existing benchmarks and our new MultiRoomR dataset.

## 1 Introduction

Multi-object rearrangement with egocentric vision in realistic simulated home environments is a fundamental challenge in embodied AI, encompassing complex tasks that require perception, planning, navigation, and manipulation. This problem becomes particularly demanding in multi-room settings with partial observability, where large parts of the environment are not visible at any given time. Such scenarios are ubiquitous in everyday life, from tidying up households to organizing groceries, making them critical for the development of next-generation home assistant robots.

Existing approaches to multi-object rearrangement typically fall into two categories: Reinforcement Learning (RL) methods and hand-coded planning systems. RL methods (Weihs et al., 2021) struggle with scaling to complex scenarios, while modular approaches that decompose tasks into subtasks (Gu et al., 2022) have different limitations. Some use predetermined skill sequences, while others employ greedy planners (Trabucco et al., 2022), restricting their ability to determine optimal object interaction orders or to handle novel problems such as blocked paths and obstructed goals. A more general approach that incorporates high-level planning would enable systems to handle these challenges without extensive retraining, particularly important for household robots operating in environments where such obstacles are common.

Although significant progress has been made in rearrangement, the majority of current research focuses on single-room settings or assumes that a large number of objects are visible at the beginning of the task, either through a third-person bird's eye view (Ghosh et al., 2022) or a first-person view where most of the room is visible (Trabucco et al., 2022). However, as we move towards the more practical version of the problems, such as cleaning a house, the majority of the objects to be manipulated are not initially visible, and existing solutions begin to falter. Rearrangement in realistic multi-room environments introduces several key challenges: 1) uncertainty over object locations, as the initial positions of objects are unknown; 2) execution efficiency of searching for objects while simultaneously moving them to the correct goal locations; 3) scalability of planning

over increasing numbers of objects and rooms; 4) extensibility to scenarios involving blocked goals or blocked paths; and 5) graceful handling of object detection failures. To enable rigorous evaluation of these challenges, we introduce MultiRoomR, a comprehensive benchmark dataset featuring diverse multi-room environments with varying degrees of partial observability, blocked paths, obstructed goals, and multiple objects distributed across rooms—scenarios that existing datasets like RoomR (Weihs et al., 2021) do not adequately represent.

To effectively tackle these complex challenges in multi-room settings, we propose a Hierarchical Object-Oriented Partially Observable Markov Decision Process (HOO-POMDP) planner that combines strategic high-level planning with specialized low-level execution. Our approach employs a high-level POMDP planner that reasons under uncertainty while leveraging object-factored belief updates, paired with a set of specialized low-level policies for executing tasks.

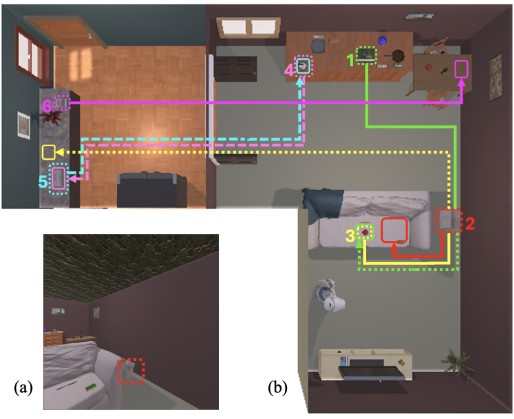

The high-level planning with uncertainty enables joint optimization over exploration and object manipulation decisions, determining when to search for unseen objects and when to rearrange detected ones based on the current belief state. Meanwhile, the low-level policies handle the execution details of navigation and manipulation, freeing the high-level planner from concerns about continuous action spaces and perceptual representations. This separation of concerns allows each component to focus on its strengths—strategic decision-making at the high level and specialized task execution at the low level. Our main contributions include:

Figure 1: a) Agent's ego-centric view at initialization. b) Top-down view of the environment showing object starting positions (dotted boxes) and goal positions (solid boxes). Lines indicate paths between start and goal states. Object 1's path is blocked by object 2, and its goal location is blocked by object 3. Object 3's path is also blocked by object 2, requiring a specific sequence: move object 2 first, then object 3, and finally object 1. Objects 4 and 5 block each other's goals, requiring one to be temporarily placed elsewhere before completing the swap.

- A hierarchical planner(HOO-POMDP) that combines an object-oriented POMDP planner with efficient belief updates and state abstraction for scalable rearrangement in multi-room environments.

- A new dataset **MultiRoomR** featuring blocked path problems and expanded room configurations alongside existing rearrangement challenges.

- An empirical evaluation of the system in an existing and the new MultiRoomR dataset in AI2Thor.

## 2 RELATED WORK

**Rearrangement:** Rearrangement is the problem of manipulating the placement of objects by picking, moving, and placing them according to a goal configuration. In this work, we are mainly concerned with the rearrangement of objects by mobile agents in simulated environments such as AI2Thor (Kolve et al., 2017) and Habitat (Szot et al., 2021) and ThreeDWorld (Gan et al., 2020). There are many versions of the rearrangement problem in literature. In tabletop rearrangement, a robot hand with a fixed base moves objects around to achieve a certain configuration in a limited space (Zhai et al., 2024; Huang et al., 2024). Many current approaches for rearrangement by mobile agents work by finding the misplaced objects and then use greedy planners to decide what order to move the objects in (Gadre et al., 2022; Trabucco et al., 2022; Sarch et al., 2022). This can lead to a high traversal cost since it is not explicitly optimized. The above works and others such as (Mirakhor et al., 2024a) are also limited to a single-room setting where most objects are visible to the agent.

Rearrangement has also been studied from the Task and Motion Planning (TAMP) perspective (Garrett et al., 2020a; 2021; 2020b). Garrett et al. (2020b) is limited to a single-room kitchen problem and assumes perfect detection of objects. Other works, such as Taskography Agia et al. (2022), focus only on the task planning part and use Learning to Plan (L4P) Silver et al. (2021), Mangannavar et al. (2025) methods to improve rearrangement. Unlike most previous work, our proposed solution optimizes the traversal cost and addresses multi-room settings and imperfect object detection in an integrated POMDP framework. Tekin et al. (2023) and Mirakhor et al. (2024b) address the multi-room rearrangement problem. However, the decision process of when to explore and when to move an object in (Tekin et al., 2023) is fixed and assumes perfect object detection. Our planner optimally combines exploration and manipulation, and naturally addresses object detector failures. Mirakhor et al. (2024b) assume that objects are always on top of or inside containers. This limits its extendability to handling new problems, such as blocked paths where the objects could be in the path of other objects and outside containers. Our approach naturally allows for these possibilities. Large language models (LLMs) have also been used to solve rearrangement (Chang et al., 2024), but the advantage of a planner is that it provides a completeness guarantee - given enough time, the planner will find a solution whereas an LLM does not provide the same guarantee.

**POMDP Planning:** Our work builds upon Wandzel et al. (2019)'s object-oriented POMDP (OO-POMDP) for 2D multi-object search, later extended to 3D environments by Zheng et al. (2023) and Zheng et al. (2022). While these OO-POMDP approaches are limited to search tasks, we extend the formulation to include rearrangement actions with corresponding belief updates. Our HOO-POMDP further introduces action abstraction, distinguishing it from existing hierarchical POMDP work (Serrano et al., 2021) which lacks object-oriented belief maintenance. This combination of hierarchical planning with object-oriented beliefs enables efficient planning for complex rearrangement tasks that would be intractable in flat POMDP representations.

## 3 Problem Formulation

**Environment and Agent**: Our agent is developed for the AI2Thor simulator environment (Kolve et al., 2017). It consists of a simulated house with a set of objects located in one or more rooms. The agent can take the following low-level actions: $A_s = (MoveAhead, MoveBack, MoveRight, MoveLeft, RotateLeft, RotateRight, LookUp, LookDown, PickObject_i, PlaceObject, Start_{loc}, Done)$. The $Move$ actions move the agent by a distance of $0.25m$. The $Rotate$ actions rotate the agent pitch by 90 degrees. The $Look$ actions rotate the agent yaw by 30 degrees. $Start$ action starts the simulator and places the agent at the given location, and the $Done$ action ends the simulation. After executing any of the actions, the simulator outputs the following information: a) RGB and Depth images, 2) the agent's position $(x, y, pitch, yaw)$, and 3) whether the action was successful. There are two types of objects in the world - *interactable* objects that can be picked and placed, and *receptacle* objects that are not movable but can hold other objects.

**Task Setup:** Rearrangement is done in 2 phases. Walkthrough phase and rearrange phase. The walkthrough phase is meant to get information about stationary objects. The 2D occupancy map is generated in this phase, as well as the corresponding 3D Map. We get the size of the house (width and length) from the environment and uniformly sample points in the environment. We then take steps to reach these locations (if possible - some might be blocked). This simple algorithm ensures we explore the full house. At each of the steps, we receive the RGB and Depth. Using this, we create a 3D point cloud at each step and combine them all to get the overall 3D point cloud of the house with stationary objects. We then discretize this point cloud into 3D map ($M^{3D}$) voxels of size 0.25m, we further flatten this 3D map into a 2D map ($M^{2D}$) of grid cells (location in the 2D map is occupied if there exists a point at that 2D location at any height in the 3D map). While doing this traversal, we also get information about the receptacles by detector on the RGB images we receive during this traversal. This ends the walkthrough phase, which needs to be done only once for any house configuration of stationary objects - walls, doors, tables, etc. Then, objects are placed at random locations (done using AI2Thor environment reinitialization). This is when the rearrangement phase begins, with the planner taking the following as input: the map generated in the walkthrough phase, the set of object classes to move, and their goal locations.

### 3.1 Rearrangement as a Object Oriented POMDP (OO-POMDP) Problem

**POMDP:** A POMDP is a 7-tuple $(S, A, T, R, \gamma, O, O_{model})$ (Kaelbling et al., 1998). The state space $S$ is the set of states in which the agent and the objects in the environment can be. Action space $A$ is

the set of actions that can be taken in the environment. The transition function $T(s, a, s') = p(s'|s, a)$ is the probability of reaching the state $s'$ when the action $a$ is taken in the current state $s$. The probability of observing $z \in O$ after having taken action $a$ in a state $s$ is defined by the observation model $O_{model}(s, a, z) = p(z|s, a)$. The reward function $R(s, a)$ defines the reward received when taking action $a$ in state $s$, and $\gamma$ is the discount factor. In a partially observed world, the agent does not know its exact state and maintains a distribution over possible states, i.e., a belief state $b$. The belief is updated when an action $a$ is taken, and observation $z$ is received with the following equation, where $\eta$ is the normalizing constant:

$$b'(s') = \eta O_{model}(s', a, z) \sum_{s \in S} T(s, a, s')b(s) \tag{1}$$

**Object Oriented POMDP:** Object-oriented POMDP factors the state and observations over the objects. Each state $s$ is represented as a tuple of its $n$ objects $s = (s_1, \ldots, s_n)$, each observation $z = (z_1, \ldots, z_n)$ and the belief state $b$ is factorized as $b = \prod_{i=0}^{n} b_i$ (Wandzel et al., 2019).

**Rearrangement as OO-POMDP:** We now instantiate the rearrangement problem as an abstract POMDP. In our definition of the abstract OOPOMDP, we make an object independence assumption - that at any given time, the observation and state of any object do not depend on any other object. More formally, $P(z_i|s_j, z_j, s_i) = P(z_i|s_i)$, observation $z_i$ is independent of the states and observations of other objects, conditioned on its own state $s_i$. Similarly, we also assume $P(s_i'|s_i, s_j, a) = P(s_i'|s_i, a)$ where $j \neq i$, i.e., the next state of object $i$ only depends on its own previous state and the action. This allows us to represent the state and observation factored by objects, which in turn helps make independent belief updates for each object (Algorithm 1).

- **State Space**: We use a factored state space that includes the robot state $s_r$, and the target object states $s_{targets}$. The complete state is represented as $s = (s_r, s_{targets})$. $s_{targets} = (s_{target_1}, \ldots, s_{target_n})$ where $n$ is the number of objects to be moved. $s_{target_i} = (loc_i, pick_i, place_{locs}, is\_held, at\_goal, g_i)$ : $loc_i$ is the current location of the object, $pick_i$ corresponds to the location from where this object can be picked, $place_{locs}$ corresponds to the set of locations (absolute 2D coordinates) from where this object can be placed, and $g_i$ is the goal location of the object. All locations are discretized grid coordinates in $M^{2D}$.

- **Action Space**: The action space consists of abstract navigation and interaction actions. $A = \{Move_{AB}, Rotate_{angle}, PickPlace_{\text{Object}_i - goal_{loc}}, Done\}$

- **Transition Model** :
  - $Move_{AB}$ - The move action moves the agent from location A to location B.
  - $Rotate_{angle}$ - The rotate action rotates the agent to a given angle.
  - $PickPlace_{\text{Object}_i - goal_{loc}}$ - The *PickPlace* action picks $Object_i$ from the current position of the robot and places it at the given $goal_{loc}$.

- **Observation Space**: We use a factored observation space similar to state space factorization. Each observation can be divided into the robot observation and object observation $z = (z_{robot}, z_{objects})$, where $z_{objects} = (z_{target_1}, \ldots z_{target_n})$. Each observation $z_{target_i} \in L \cup \text{Null}$ - is a detection of the object $i$'s location or *Null* based on the detector's output for object $i$ ( $L$ is the set of all possible locations in $M^{2D}$).

- **Observation model:** By definition of $z$ above, $\Pr(z|s) = \Pr(z_r|s_r) \Pr(z_{\text{objects}}|s_{targets})$ and $\Pr(z_r|s_r) = 1$ since the robot pose changes deterministically. Under the conditional independence assumption, $\Pr(z_{\text{objects}}|s)$ can be compactly factored as follows:

$$\Pr(z_{\text{objects}}|s) = \Pr(z_{target_1}, \ldots, z_{target_n}|s_{target_1}, \ldots s_{target_n}, s_r) \tag{2}$$

$$= \prod_{i=1}^{n} \Pr(z_{target_i}|s_{target_1}, \ldots, s_{target_n}, s_r) \quad \text{(all } z_{target_i} \text{ are independent)} \tag{3}$$

$$= \prod_{i=1}^{n} \Pr(z_{target_i}|s_{target_i}, s_r) \quad (z_{target_i} \text{ does not depend on state of other objects}) \tag{4}$$

$\Pr(z_i|s_{target_i}, s_r)$ is defined differently for each object based on the object detector's capability to detect the object of interest and the current state. More details are in A.1.2.

- **Reward Function**:
  - $Move_{AB}$ : The cost of moving from location A to B [$Cost = -1 * N_a$ (where $N_a$ number of required actions)].

- *Rotate$_{angle}$*: The cost of rotating the agent from the current rotation to the final given angle.
- *PickPlace$_{Object_i - goal_{loc}}$*: Cost of moving from current location to goal location + cost of pick + cost of place. It gets an additional reward of $50$ if the object is being placed at its goal location $g_i$ and this $g_i$ is free in the current state.
- *Done* - This action receives a reward of $50$ if all objects have been placed at their goal location and $-50$ otherwise.

## 4 HIERARCHICAL OBJECT ORIENTED POMDP (HOO-POMDP) PLANNING

This section presents our hierarchical planning solution designed to solve multi-object rearrangement problems in partially observable, multi-room simulated home environments.

**Algorithm 1:** HOO-POMDP Planner

1. $env \leftarrow$ INITIALIZEENV() ;
2. $agent \leftarrow$ INITIALIZEAGENT()
3. $belief \leftarrow$ INITIALIZEBELIEFSTATE()
4. $loc \leftarrow$ RANDOM()
5. $lowLevelAct \leftarrow$ START$_{loc}$
6. **while** NOT TASKCOMPLETE() **do**
7.  $rgb, depth \leftarrow env.$EXECUTE($lowLevelAct$)
8.  $obs \leftarrow$ PERCEPTIONSYSTEM($rgb, depth$)
9.  $belief \leftarrow$ BELIEFUPDATE($belief, obs, lowLevelAct$)
10.  $absState \leftarrow$ GENERATEABSSTATE($belief$)
11.  $absAct \leftarrow$ POUCTPLANNER($absState, belief$)
12.  **if** $absAct =$ DONE **then**
13.   **return**
14.  $lowLevelPolicy \leftarrow$ GETLOWLEVELPOL-ICY($absAct$)
15.  $lowLevelAct \leftarrow lowLevelPolicy.$GETACT($absAct, rgb, depth$)

**Algorithm 2:** BELIEFUPDATE

1. **Input:** beliefState $b$, observation $z$, action $a$
2. **for** each object $i$ in $b$ **do**
3.  **for** each possible state $s_{ij}$ of object $i$ **do**
4.   **if** action $\in$ {Pick, Place} and successful **then**
5.    **if** action is pick **then**
6.
$$b'_i(s_{ij}) \leftarrow \begin{cases} 1 & \text{if } s_i = \text{action.agentLocation} \\ 0 & \text{otherwise} \end{cases}$$
7.    **if** action is place **then**
8.
$$b'_i(s_{ij}) \leftarrow \begin{cases} 1 & \text{if } s_{ij} = \text{action.goalLocation} \\ 0 & \text{otherwise} \end{cases}$$
9.   **else**
10.    $b'_i(s_{ij}) \leftarrow p(z_i|s_{ij})b_i(s_{ij})$
11.   **end if**
12.  **end for**
13. **end for**
14. **return** $b'$

**Overview:** Once the initial list of receptacles and $M^{3D}$ have been generated, they, along with the goal information, are sent to the HOO-POMDP planner. The system operates in a cyclic fashion, integrating *perception*, *belief update*, *state abstraction*, *abstract planning*, and *action execution* (see Figure 2 and Algorithm 1). First, the *perception system* detects objects in the RGB and depth image and outputs the observation $z$, which is used by the *belief update system* to update its belief state. The belief state consists of the probability of each object being at a certain location in $M^{2D}$. The *abstraction system* uses this information to update its abstract state. The updated abstract state is sent to the abstract POMDP planner, which outputs a sub-goal that corresponds to a low-level policy. The low-level *policy executor* executes the low-level policy corresponding to the sub-goal.

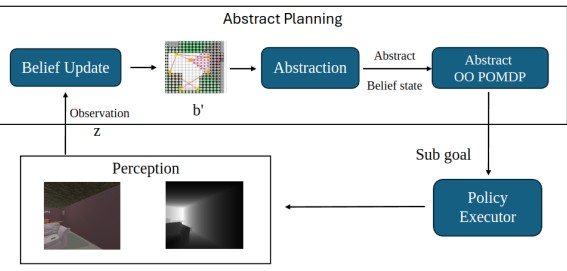

Figure 2: The agent receives RGB and depth images from environment at the start. The vision module creates the observation from this input and sends it to the belief update system. Belief is updated based on observation, and an abstract state is generated, which is sent to the OO POMDP Planner that outputs sub-goals. The sub-goals are used by the low-level policy executors to plan and execute low-level actions in the environment.

This might involve navigating to a specific location, grasping an object, or placing an object in a new position. After each action is executed, the environment state changes. The agent receives new output from the environment, and the cycle repeats until the overall rearrangement task is completed. In the rest of this section, we will discuss each of the subsystems and their interaction.

**Abstract OOPOMDP Planner**: Given a task defined as an abstract OOPOMDP and an initial abstract state, we use partially observable UCT (PO-UCT) (Silver & Veness, 2010) to search through

the space of abstract actions to find the best sub-goal. POUCT extends the UCT algorithm Kocsis & Szepesvári (2006) to partially observable settings, building a search tree over histories rather than states, where a history is a sequence of actions and observations $h_t = (a_1, z_1, \ldots, a_t, z_t)$. For each history node $T(h)$, the algorithm maintains a count variable $N(h)$ and a value variable $v(h)$ representing visit frequency and expected value. The algorithm samples a state from the belief space $b$ and selects the action with the best value using $V(ha) = V(ha) + c\sqrt{\frac{\log N(h)}{N(ha)}}$ when all child nodes exist. Otherwise, it uses a random rollout policy for simulations before updating the tree and selecting the best action. The full algorithm appears in Algorithm 3 in the Appendix.

In our HOO-POMDP planner, abstract actions are initialized based on the abstract state for each object $s_i = (loc_i,\ pick_i,\ place_{locs},\ is\_held,\ at\_goal)$. A separate $Move_{AB}$ is initialized with $A = agent\_pos$ and $B = $ all pick locations defined for all objects. $Rotate_{angle}$ - for all objects, less than 2m from the agent, the angle is computed based on the agent's required orientation to view the object from its current position. $PickPlace$ - is defined for each object where the agent is less than 2m away from that particular object, for all locations in the $place_{locs}$ as $goal_{loc}$ initializing a set of $PickPlace$ actions for each object.

**Low-Level Policy Executor:** Each sub-goal (instantiated abstract action) output by the abstract planner corresponds to a policy. When the planner outputs a sub-goal, the information in the sub-goal is used to initialize the low-level policy. The output from the low-level policy is a sequence of low-level actions. We then execute the first low-level action in this sequence in the environment.

The *Move* sub-goal corresponds to the *Move* policy, which uses the $A^*$ algorithm to move from location A to B. The *Rotate* policy also uses the $A^*$ algorithm. The *PickPlace* policy consists of 2 RL agents and $A^*$ that picks the object from the current location and places it at the goal location.

- Sub-goal $Move_{AB}$ gives the *Move* policy the location B to move to from location $A$, which is used to initialize the $A^*$ algorithm and get a sequence of low-level move actions to reach goal location $B$. The action space available to the system is all the *Move* actions and all the rotate actions. It uses an Euclidian distance-based heuristic.

- Sub-goal $Rotate_{angle}$ gives the *Rotate* policy the final angle to be at, which is used as the final state the $A^*$ system must reach. $A^*$ outputs a sequence of rotate actions.

- Sub-goal $PickPlace_{Object_i - goal_{loc}}$ provides the object to interact with and which location to place it at. The policy takes this information as input and outputs a sequence of actions consisting of *Pick*, *Place*, and navigation actions. The *PickPlace* consists of 3 separate components a) An RL model trained to pick an object, b) the $A^*$ navigation model to go its destination c) An RL model trained to place the object when the agent is near the goal. All 3 of these run sequentially and make up the *PickPlace* Policy. It is designed this way to improve modularity and reduce the complexity of each part. All details of the RL policy training are described in appendix A.5

**Perception System:** Once the agent executes the low-level action, it receives an RGB and Depth image. A detector is used to detect objects in the RGB image, and the depth map is used to get their $3D$ location in the world. This is used to generate the object-oriented observation $z = (z_1, \ldots, z_n)$.

**Belief Update**: Algorithm 2 presents the belief update function for our HOO-POMDP. The UpdateBelief function takes as input the current belief state $b$, the performed action $a$, and the received observation $z$ (Line 1.). For each object $i$ in the belief state and each possible state $s_{ij}$ ($j = 1, \ldots, L$, where $L$ is the set of all its possible locations in $M^{2D}$) of that object, the algorithm updates the belief based on the action type and its success status. For successful 'pick' and 'place' actions (Line 4.), the belief update is deterministic. When a 'pick' action succeeds, it assigns a belief of **1** if the object's state corresponds to the agent's location and **0** otherwise (Line 6.). For successful 'place' actions, it sets the belief to **1** if the object's state matches the action's goal location, and **0** otherwise (Line 8.). For navigation actions or when 'pick'/'place' actions fail, the algorithm applies a probabilistic update using the observation model $p(z_i|s_{ij})$ (details in Appendix A.1.2) and the prior belief $b_i(s_{ij})$ (Line 10.).

**Generating Abstract State:** We now have a belief state over the set of all possible locations for each object. We need to generate the abstract object-wise state consisting of object location information and their corresponding pick-and-place information. The information that needs to be computed for each object is as follows: $pick_i,\ place_{locs},\ is\_held,\ at\_goal$.

The value for $is\_held$ comes from the previous low-level action and previous state. If the previous state had $is\_held$ as false and low-level action was to pick the object of interest, $is\_held$ is set to true. If the previous action was not a pick or a pick action for a different object, then the variable remains unchanged. If the previous action was $place$ and $is\_held$ is true, then it is set to False.

The value for $at\_goal$ is copied from the previous state if the last low-level action was not the place action. If it was, and if $is\_held$ was true in the previous state, then $at\_goal$ is set to true.

The values for $place_{locs}$ are sampled from the object goal location and three nearby receptacles as alternate goal locations for the object. For each of these goal locations, a location from where the object can be placed is sampled.

The location $pick_i$ is sampled based on the belief distribution of where the object could be. It is the location from which the object can be picked. If the distribution over location is spread out, we sample multiple locations (by ensuring each sampled location is far from the other sampled locations for the same object). For both the locations in $place_{locs}$ and for the location $pick_i$, we then check if they are reachable. If they are not, those locations are discarded.

This sampling method enables our system to handle scenarios involving blocked goals, object swaps, and blocked paths effectively. If an object's path is blocked, the planner will receive information indicating that there is no accessible location from which to pick up the object, necessitating the relocation of other objects first. When placing objects, we provide alternative receptacle locations. This approach allows us to move an object to another location if its goal position is blocked, thereby freeing up its current location. This strategy addresses both blocked goal and swap scenarios. Furthermore, this sampling process enhances our system's extensibility. We can incorporate additional constraints based on new object properties. For example, if opening an object requires interaction from a distance, the sampler can ensure that the sampled location is sufficiently far to enable successful opening. After creating this abstract state, it is sent to the abstract planner, and the cycle starts again.

## 5 EXPERIMENTS

### 5.1 DATASETS

- **RoomR:** This is the rearrangement challenge dataset proposed by Batra et al. (2020). It contains single-room environments with 5 objects to be rearranged. It has 25 room configurations with 40 different rearrangements for each room configuration.

- **ProcTHORRearrangement (Proc):** This is a dataset present in AI2Thor, which is bigger in terms of the rooms (two rooms, five objects) and, hence, partial observability. It has 125 room configurations with 80 rearrangements for each room configuration.

- **Multi RoomR:** We introduce a novel dataset designed to address more challenging problems, featuring larger environments (2-4 rooms) and an increased number of objects (10-20 objects). It has 400 room configurations. More details in Appendix A.2.5.

### 5.2 METRICS

- **Scene Success (SS): 1** if all objects have been moved to the correct goal locations, **0** otherwise.

- **Object Success (OS):** 100* (Total Objects successfully moved)/(Total objects to move) - this metric captures the percentage of objects moved to the correct goal location.

- **Total Actions taken (TA):** The average number (rounded up) of actions taken during successful runs where the scene was fully rearranged, which is a measure of the efficiency of the system.

### 5.3 METHODS AND BASELINES DEFINITION

- **HOOP (HOO-POMDP Planner):** Our proposed solution. In this, we will solve the rearrangement challenge where the agent handles perception uncertainty (the detector fails to detect objects in the visual field) along with the object's position uncertainty using the proposed HOO-POMDP planner.

- **Frontier Exploration + Hand-Coded Interaction (FHC) :** This baseline employs a frontier exploration strategy that systematically explores the environment until objects that need rearrange-

Table 1: Comparison between all methods. The difficulty is represented in terms of the following: a) **#BP**: Number of objects blocking the path that need to be moved out of the way, b) **#Rm**: Number of rooms in the environment, c) **#V**: Number of objects initially visible. NA: Not applicable as no scene was fully rearranged. NC - Not computable as the method cannot handle blocked paths

| Dataset | Objs | #BP | #Rm | #V | Methods | | | | | | | | | | | | Ablation | | | Oracle Settings | | | | | |
| | | | | | HOOP | | | FHC | | | VRR | | | MSS | | | HOOP-HP | | | PK | | | PD | | |
| | | | | | SS↑ | OS↑ | TA↓ | SS↑ | OS↑ | TA↓ | SS↑ | OS↑ | TA↓ | SS↑ | OS↑ | TA↓ | SS↑ | OS↑ | TA↓ | SS↑ | OS↑ | TA↓ | SS↑ | OS↑ | TA↓ |
|---|---|---|---|---|---|---|---|---|---|---|---|---|---|---|---|---|---|---|---|---|---|---|---|---|---|
| RoomR | 5 | 0 | 1 | 3-4 | **49** | **71** | **211** | 38 | 58 | 269 | 7 | 31 | 256 | 21 | 44 | 267 | 13 | 33 | 302 | 63 | 88 | 176 | 62 | 87 | 189 |
| Proc | 5 | 0 | 2 | 2-3 | **46** | **68** | **352** | 32 | 61 | 411 | 2 | 19 | 382 | 14 | 29 | 395 | 9 | 29 | 410 | 60 | 82 | 203 | 60 | 81 | 269 |
| Multi RoomR | 10 | 0 | 2 | 2-3 | **32** | **65** | **710** | 20 | 44 | 931 | 0 | 13 | NA | 8 | 25 | 920 | 5 | 25 | 1029 | 41 | 78 | 457 | 40 | 78 | 529 |
| | 10 | 1 | 2 | 2-3 | **21** | **49** | **789** | 12 | 38 | 993 | 0 | 9 | NA | NC | NC | NC | 2 | 19 | 1092 | 33 | 69 | 489 | 29 | 67 | 587 |
| | 10 | 0 | 3-4 | 1-2 | **30** | **62** | **1189** | 19 | 34 | 1345 | 0 | 8 | NA | 0 | 14 | NA | 3 | 16 | 1408 | 39 | 75 | 726 | 37 | 74 | 834 |
| | 10 | 1 | 3-4 | 1-2 | **18** | **44** | **1321** | 9 | 26 | 1490 | 0 | 5 | NA | NC | NC | NC | 1 | 7 | 1549 | 32 | 70 | 789 | 31 | 70 | 985 |
| | 15 | 0 | 3-4 | 2-3 | **22** | **59** | **1228** | 12 | 31 | 1605 | 0 | 9 | NA | 0 | 11 | NA | 0 | 5 | NA | 32 | 78 | 895 | 30 | 74 | 921 |
| | 15 | 1 | 3-4 | 2-3 | **14** | **41** | **1416** | 7 | 23 | 1886 | 0 | 5 | NA | NC | NC | NC | 0 | 6 | NA | 29 | 71 | 988 | 25 | 69 | 965 |
| | 20 | 0 | 3-4 | 2-4 | **17** | **55** | **1621** | 0 | 18 | NA | 0 | 6 | NA | 0 | 9 | NA | 0 | 5 | NA | 27 | 75 | 1168 | 27 | 74 | 1197 |
| | 20 | 1 | 3-4 | 2-4 | **10** | **36** | **1786** | 0 | 11 | NA | 0 | 4 | NA | NC | NC | NC | 0 | 4 | NA | 22 | 70 | 1307 | 20 | 68 | 1336 |

ment are discovered. It uses a confidence-based approach where an object is considered detected when the belief probability exceeds 70%. Once detected, the object is immediately rearranged, and the system resumes exploration for the next undetected object. More details in appendix A.6

- **VRR**: This is the model from Weihs et al. (2021) trains and RL agent using PPO (Schulman et al., 2017) and imitation learning with RGB images as input, builds a semantic map using Active Neural Slam (Chaplot et al., 2020) and outputs low-level actions directly.

- **MSS** : System from Trabucco et al. (2022) first builds a 3D map of the whole world, then navigates to each object that needs to be moved and rearranges it.

- **HOOP-HP** (ablation): In this ablation setting, we remove the hierarchical planning and use the POMDP planner to output low-level actions directly.

- **Perfect Knowledge (PK):** In this oracle setting of our system, we will start with all the information about the world. That is, we know the initial locations of all the objects. This is the upper limit of the system's performance, as there is no uncertainty to manage.

- **Perfect Detector with partial observability (PD):** In this oracle setting of our system, we solve our multi-object rearrangement problem with a perfect detector (objects in the visual field are detected with 100% probability). The challenge is to find all objects and move them around efficiently.

**Experimental setup:** Each experimental setting was evaluated across 100 distinct rearrangement configurations. For the RoomR dataset, we utilized 25 different room setups, with four rearrangement configurations per setup. For the other datasets, we employed 100 unique room configurations, each with one rearrangement configuration. In the blocked path scenario, all scenes contained a minimum of one object that needed to be moved to its goal position from its initial location to enable the rearrangement of other objects. Results for blocked goal and swap cases in table 3 in the appendix.

## 6 RESULTS AND DISCUSSION

**Methods comparison:** PK represents our agent's upper performance bound with no uncertainty. As shown in Table 1, the PD setting achieves comparable scene success and object success rates to PK, demonstrating our POMDP planner effectively handles partial observability. The primary difference appears in action count—PD requires more steps than PK due to PD actually needing to look for the objects and plan with this uncertainty. HOOP system with imperfect detection and partial observability shows reduced success rates as the planner must account for detection failures. Despite only 50-60% detection success across object classes, HOOP still solves many problems comparable to PD, demonstrating its effectiveness in handling detector failures. The increased exploration necessitated by detector failures results in more steps than other methods. The substantial performance gap between HOOP and HOOP-HP demonstrates the critical importance of our hierarchical abstraction approach.

**Comparison to baselines :** Our system significantly outperforms all baselines, demonstrating the clear advantages of a principled planning approach. While VRR's pure reinforcement learning strategy struggles to scale beyond simple environments, failing dramatically as complexity increases, both hand-coded methods (FHC and MSS) also show substantial limitations. FHC not only struggles with complex spatial reasoning but also fails to handle object detection failures effectively, as it lacks a belief-based framework to account for perceptual uncertainty. MSS's rigid explore-then-rearrange strategy prevents it from addressing blocked paths and causes inefficient execution even in simpler scenarios. These results highlight that planning-based approaches provide essential flexibility and robustness for real-world rearrangement tasks where uncertainty and complexity are inevitable, outperforming both end-to-end learning and hand-engineered heuristic methods.

It is important to note that our system addresses a variant of the multi-object rearrangement problem that differs in key aspects from those tackled by existing baselines. The primary distinction is that we are given information about the classes of objects to be moved, whereas other systems, VRR (Weihs et al., 2021) and MSS (Trabucco et al., 2022)operate without this knowledge. On the other hand, our problem formulation introduces its own set of challenges. In particular, while existing systems report initial visibility of approximately 60% ((Mirakhor et al., 2024b), Table 1) of target objects at the outset of their tasks, only about 20% of the objects are initially visible in our problem settings in MultiRoomR, necessitating more extensive and strategic exploration. This reduced initial visibility significantly increases the complexity of our task in terms of efficient exploration and belief management, and underscores the effectiveness of our approach.

**Comparison across datasets :** Our system maintains relatively consistent performance across datasets when controlling for room and object count. The RoomR (single room) and Proc (two rooms) datasets show similar success rates despite differing room counts. As we scale to MultiRoomR with more objects, the object success rate decreases gradually, while the scene success drops more sharply since it requires all objects to be successfully rearranged.

**Performance across challenges :** The blocked path variants consistently show lower success rates due to the increased complexity and limitations in our low-level manipulation policy, which struggles more with floor objects than tabletop ones. The required PickPlace actions increase for blocked path scenarios as the agent must execute intermediate movements to clear pathways.

**Error analysis :** Most failures stem from two factors: low-level policy limitations in pick/place actions and belief estimation errors from detector failures. When the detector misses an object multiple times, our belief about its presence decreases, making the planner unlikely to revisit that area. False positives can also cause manipulation of incorrect objects. Despite these challenges, our approach significantly outperforms baselines across diverse scenarios.

**Limitations:** Our system's object independence assumption fails in cluttered environments, where object states and observations are influenced by nearby objects. This requires modifying our object-oriented belief update to handle these interactions. HOO-POMDP also cannot handle an unknown class of objects. While it is easy enough to categorize all such objects into a single 'unknown' class, the difficult part is to plan to find an empty space to move the unknown object to.

## 7 CONCLUSION AND FUTURE WORK

In this paper, we presented a novel Hierarchical Object-Oriented POMDP Planner (HOO-POMDP) for solving multi-object rearrangement problems in partially observable, multi-room environments. Our approach decomposes the complex task into a high-level abstract POMDP planner for generating sub-goals and low-level policies for execution. Key components include an object-oriented state representation, a belief update that handles perception uncertainty, and an abstraction system that bridges the gap between continuous and discrete planning. Experimental results across multiple datasets demonstrate the effectiveness of our approach in handling challenging scenarios such as blocked paths and goals. HOO-POMDP showed robust performance in terms of success rate and efficiency comparable to oracle baselines with perfect knowledge or perfect detection. Notably, our method scaled well to environments with more objects and rooms. Future work could partially relax our object independence assumption to better reflect real-world scenarios and local object dependencies. Another key direction for future work is deploying our planner on physical robots by replacing the reinforcement learning controller with a low-level robot controller.

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

# A APPENDIX

## A.1 OBJECT DETECTION

### A.1.1 DETECTION MODEL : YOLOV10

We collect data from the AI2Thor simulator. We do this by placing the agent in random locations in 500 scenes and extracting the RGB images along with the ground truth object bounding box annotations from the simulator. We have 50 pickupable object classes in all our scenes combined. We train the YoloV10 detector (YoloV10-Large, with 25 million parameters) on 10,000 images collected from these 500 scenes. It is trained for 500 epochs, batch size 16. It is trained on RTX 3080 for 12 hours.

### A.1.2 OBSERVATION PROBABILITY FOR POMDP

The probability of each individual observation based on the current state is the following.

$$
\Pr(z_i|s_{target_i}, s_{\text{robot}}) = \begin{cases} 1.0 - \text{TP} & s_{target_i} \in \mathcal{V}(s_{\text{robot}}) \land z_i = \text{null} \\ \delta\text{FP}/|\mathcal{V}_E(r)| & s_{target_i} \in \mathcal{V}(s_{\text{robot}}) \land \|z_i - s_{target_i}\| > 3\sigma \\ \delta * TP & s_{target_i} \in \mathcal{V}(s_{\text{robot}}) \land \|z_i - s_{target_i}\| \leq 3\sigma \\ 1.0 - \text{FP} & s_{target_i} \notin \mathcal{V}(s_{\text{robot}}) \land z_i = \text{null} \\ \delta\text{FP}/|\mathcal{V}_E(r)| & s_{target_i} \notin \mathcal{V}(s_{\text{robot}}) \land z_i \neq \text{null} \end{cases}
$$

The detection model is parameterized by

- TP: Is the True positive of the Detection model for object class $i$.
- FP: Is the False positive of the Detection model for object class $i$.
- $r$: is the average distance between the agent and the object for true positive detections.
- $V_E(r)$: It is the visual field of view of 90 degrees within distance $r$.
- $\delta$ : is the distance weight, it is 1 if detection is within $V_E(r)$, else $\delta = 1/d$, where $d$ is the distance from the robot to the object.

The list of $TP$, $FP$, and $r$ for the object classes in the dataset is presented in table 2.

## A.2 MULTIROOMR DATASET DETAILS

### A.2.1 OVERVIEW

Recent advancements in robotic rearrangement have been facilitated by datasets such as RoomR Batra et al. (2020) and ProcThorRearrangement. However, these datasets exhibit significant limitations that prevent them from capturing the complexity of real-world rearrangement scenarios. Specifically, existing datasets are constrained in their scope, typically featuring only 5 objects and limited room configurations (single room in RoomR, two rooms in ProcThorRearrangement). These constraints fail to represent the challenges inherent in real-world environments, where rearrangement tasks frequently span multiple rooms and involve numerous objects.

To address these limitations, we present MultiRoomR, a comprehensive dataset designed to bridge three critical gaps in existing benchmarks. First, MultiRoomR emphasizes partial observability by incorporating scenes with 2-4 rooms where most objects are not immediately visible, thereby ensuring that systems must develop robust strategies for handling incomplete information. Second, the dataset increases scene complexity by including 10-20 objects per environment, necessitating the development of efficient and scalable solutions that can handle larger object sets without compromising optimality. Third, MultiRoomR introduces realistic constraints such as blocked paths, with 50% of scenes containing at least one object that obstructs access to other objects. This feature specifically tests a system's ability to reason about sequential manipulation, as encountered in practical scenarios where intermediate object movements are necessary to complete the primary rearrangement task. By incorporating these challenging aspects, MultiRoomR provides a more rigorous evaluation framework that better aligns with real-world requirements. Systems that demonstrate strong performance on this

Table 2: Performance Metrics by Class : TP = True Positive, FP = False positive, r = average distance

| Class | r (m) | TP | FP |
|---|---|---|---|
| AlarmClock | 3.010 | 0.383 | 0.022 |
| Apple | 3.298 | 0.065 | 0.002 |
| BaseballBat | 2.941 | 0.499 | 0.011 |
| BasketBall | 2.631 | 0.336 | 0.003 |
| Book | 2.888 | 0.535 | 0.101 |
| Bottle | 2.733 | 0.465 | 0.006 |
| Bowl | 2.695 | 0.448 | 0.073 |
| Box | 3.977 | 0.225 | 0.012 |
| Bread | 1.523 | 0.082 | 0.010 |
| ButterKnife | 2.084 | 0.156 | 0.009 |
| CD | 2.082 | 0.085 | 0.001 |
| Candle | 3.325 | 0.048 | 0.004 |
| CellPhone | 2.137 | 0.327 | 0.006 |
| CreditCard | 1.107 | 0.042 | 0.002 |
| Cup | 2.505 | 0.513 | 0.012 |
| DishSponge | 1.714 | 0.306 | 0.003 |
| Kettle | 2.655 | 0.415 | 0.001 |
| KeyChain | 1.725 | 0.154 | 0.007 |
| Knife | 1.226 | 0.056 | 0.002 |
| Ladle | 2.333 | 0.015 | 0.000 |
| Laptop | 3.405 | 0.605 | 0.019 |
| Lettuce | 2.681 | 0.336 | 0.003 |
| Mug | 2.734 | 0.529 | 0.010 |
| Newspaper | 2.286 | 0.264 | 0.005 |
| Pan | 2.757 | 0.350 | 0.012 |
| PaperTowelRoll | 3.066 | 0.338 | 0.018 |
| Pen | 2.471 | 0.081 | 0.013 |
| Pencil | 1.853 | 0.040 | 0.015 |
| PepperShaker | 2.042 | 0.310 | 0.016 |
| Pillow | 3.615 | 0.683 | 0.037 |
| Plate | 2.278 | 0.355 | 0.010 |
| Plunger | 2.900 | 0.745 | 0.005 |
| Pot | 4.064 | 0.417 | 0.010 |
| Potato | 2.138 | 0.157 | 0.004 |
| RemoteControl | 2.176 | 0.324 | 0.025 |
| SaltShaker | 1.940 | 0.098 | 0.010 |
| SoapBottle | 3.147 | 0.519 | 0.038 |
| Spatula | 1.443 | 0.134 | 0.002 |
| SprayBottle | 2.744 | 0.085 | 0.031 |
| Statue | 3.095 | 0.650 | 0.033 |
| TeddyBear | 3.093 | 0.417 | 0.003 |
| TennisRacket | 3.111 | 0.128 | 0.017 |
| TissueBox | 4.087 | 0.286 | 0.003 |
| ToiletPaper | 2.806 | 0.383 | 0.006 |
| Vase | 3.230 | 0.699 | 0.095 |
| Watch | 1.661 | 0.210 | 0.006 |
| WineBottle | 2.903 | 0.667 | 0.003 |

dataset are likely to be more suitable for deployment in actual home environments, where partial observability, multi-object manipulation, and complex spatial reasoning are commonplace challenges. We also provide the dataset [1] and code [2] for the community to use. The dataset consists of 400 distinct room configurations, with varying complexity in terms of room count and object arrangements. The dataset composition and room configurations are presented in the rest of this section.

---

[1] Dataset submitted as part of supplementary material

[2] Link to code: `https://anonymous.4open.science/r/ICML_POMDP_rearrangement-4460/`

### A.2.2 DATASET CREATION CRITERIA

Data generation has 2 parts:

- Basic scene generation is done using ProcThor (Deitke et al., 2022). To increase task complexity, we (1) add the extra condition, the average distance between an object and its goal is above 25 steps, and (2) sample goal positions in different rooms for at least 50% of the objects.
- Second, for generating blocked path scenes, we use the connected components algorithm. We construct a connected graph of all 2D navigable points in the room. We then find 2*2 grid locations that, if not navigable, will make a portion of the house not reachable. Among these possible options, we pick the grid that blocks the maximum number of objects from the current starting location of the agent to ensure that not moving a blocking object (of size 2*2) renders a large number of objects inaccessible. We will add this in the appendix.

### A.2.3 DATASET COMPOSITION

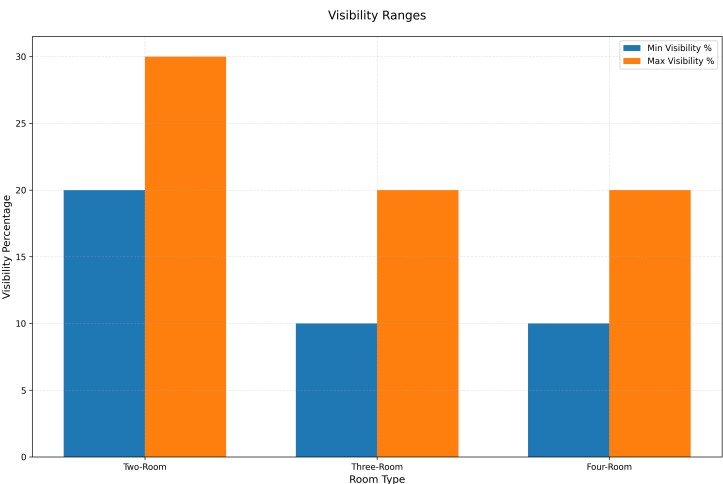

Figure 3: Range of percentage of objects visible at the beginning of a scene

- **Total Size:** 400 room configurations.
- **Object Types:** Comprehensive selection from AI2Thor environment (see Table 2)
- **Object Selection Criteria:** Includes the majority of AI2Thor objects, excluding objects too small for reliable detection even at close range.
- Figure 3 shows the range of the percentage of objects visible at the beginning of the scene. The lower the objects visible, the more exploration needs to be done thus making the problem harder. It ranges between 10-40% in our dataset, where it is 60% in Mirakhor et al. (2024b) Table 1. This implies, only 20-30% of objects are visible in the beginning for two-room scenes and so on for other room settings.

### A.2.4 ROOM CONFIGURATION DISTRIBUTION

*Two-Room Configurations*

- Total configurations: 200
- Objects per configuration: 10
- Initial visibility: 20-30% of objects visible at the beginning of the scene

- Path characteristics:

  - 50% contain blocked paths.
  - 10 rearrangements per configuration.

*Three-Room Configurations*

- Total configurations: 100
- Path characteristics:

  - 50% contain blocked paths.
  - 30 rearrangements per configuration.
  - Distribution: 10 rearrangements each for 10, 15, and 20 objects.
  - Initial visibility: 10-20% of objects visible at the beginning of the scene

*Four-Room Configurations*

- Total configurations: 100
- Path characteristics:

  - 50% contain blocked paths.
  - 30 rearrangements per configuration.
  - Distribution: 10 rearrangements each for 10, 15, and 20 objects.
  - Initial visibility: 10-20% of objects visible at the beginning of the scene

### A.2.5 OBJECT PLACEMENT CRITERIA

1. **Room-wide Movement Requirement:** Each room must contain at least one object requiring movement, ensuring comprehensive exploration by the agent.
2. **Blocking and Swapping Scenarios:** Configurations include:

   - Objects blocking goal locations of other objects.
   - Objects mutually blocking each other's goals (swap cases).

3. **Path Blocking Optimization:** In scenes with blocked paths, blocking objects are strategically placed to maximize inaccessible house area such that at least one object must be moved out of the way to access all objects.

## A.3 POUCT

## A.4 QUALITATIVE RESULTS

We test our system with different depths for the MCTS planner to see how much look-ahead affects the performance of our system. MCTS search depth is one of the important factors determining the amount of exploration and the time taken for search at each step. Results are shown in table 3. The depth for the system - HOOP is 12, and the depth for HOOP-MCTS_1 is 1. From the table, we can see that the greedy approach of just looking ahead by 1 step is not enough to solve the rearrangement problem. This is because, when we look ahead only 1 step, we do not get enough reward/feedback from the world about what is a good path to take, and hence makes it extremely hard to solve the problem. We can also see that, in the problems it does solve, it takes significantly longer to solve.

We have run our system on the RoomR dataset, and we have results from Mirakhor et al. (2024a) and Mirakhor et al. (2024b) on the RoomR dataset Batra et al. (2020) and show the results in table 4 We can see that our success rate is slightly higher but this is with the caveat that we use more information at the beginning of rearrangement about what objects need to be moved.

---

**Algorithm 3:** POUCT Planner

---

1. **Input:** abstractState $abs$, beliefState $b$
2. $T \leftarrow \{\}$
3. **For** $j = 0$ to $SIMULATIONS$:
4.     $\hat{s} \leftarrow$ SAMPLE($b$)
5.     SIMULATE($\hat{s}, \{\}, 0, abs$)
6. **Return** $\arg\max_a V(ha)$
7. **Function** SAMPLE($b$):
8.     **For** each object $o$:
9.       Sample $\hat{s}_o \sim b_o$
10.     **Return** $\bigcup \hat{s}_o$
11. **Function** SIMULATE($s, h, depth, abs$):
12.     **if** $\gamma^{depth} < \epsilon$: **return** 0
13.     **if** $h \notin T$:
14.       Initialize $T(ha)$ for all actions $a$
15.       **Return** ROLLOUT($s, h, depth, abs$)
16.     $a \leftarrow$ selectMaxAction()
17.     $(s', z, r) \sim \mathcal{G}(s, a)$
18.     $R \leftarrow r + \gamma \cdot$ SIMULATE($s', hao, depth + 1$)
19.     Update $T(ha)$ with new value and count
20.     **Return** $R$
21. **Function** ROLLOUT($s, h, depth, abs$):
22.     **if** $\gamma^{depth} < \epsilon$: **return** 0
23.     $a \sim \pi_{rollout}(h, abs)$
24.     $(s', o, r) \sim \mathcal{G}(s, a)$
25.     **Return** $r + \gamma \cdot$ ROLLOUT($s', hao, depth + 1$)

---

Table 3: Performance metrics for our method across different depths for MCTS search. Metrics: Success Score (SS), Object Success Rate (OSR), Task Actions (TA), execution Time in minutes, and number of objects initially visible (#V). The difficulty parameters include the number of blocked goals (#BG), objects to be swapped (#Sw), blocking objects (#BP), and number of rooms (#Rm). NA: Not Applicable as no scenes were fully rearranged.

| Dataset | Objs | #BG | #Sw | #BP | #Rm | #V | | SS↑ | OSR↑ | TA↓ | Time(m)↓ | | SS↑ | OSR↑ | TA↓ |
|---|---|---|---|---|---|---|---|---|---|---|---|---|---|---|---|
| | | ap | | | | | | | HOOP | | | | | HOOP-MCSTS_1 | |
| RoomR | 5 | 1 | 0 | 0 | 1 | 3-4 | | **49** | **71** | **211** | 1.61 | | 8 | 26 | 565 |
| Proc | 5 | 1 | 0 | 0 | 2 | 2-3 | | **46** | **68** | **352** | 3.42 | | 2 | 12 | 875 |
| | 10 | 1 | 1 | 0 | 2 | 2-3 | | **32** | **65** | **710** | 7.89 | | 0 | 7 | NA |
| | | 2 | 1 | 1 | 2 | 2-3 | | **21** | **49** | **789** | 8.98 | | 0 | 3 | NA |
| Multi RoomR | 10 | 2 | 1 | 0 | 3-4 | 1-2 | | **30** | **62** | **1189** | 13.45 | | 0 | 5 | NA |
| | | 2 | 1 | 1 | 3-4 | 1-2 | | **18** | **44** | **1321** | 15.97 | | 0 | 2 | NA |
| | 15 | 1 | 1 | 0 | 3-4 | 2-3 | | **22** | **59** | **1228** | 19.89 | | 0 | 0 | NA |
| | | 2 | 1 | 1 | 3-4 | 2-3 | | **14** | **41** | **1416** | 22.12 | | 0 | 0 | NA |
| | 20 | 2 | 1 | 0 | 3-4 | 2-4 | | **17** | **55** | **1621** | 27.61 | | 0 | 0 | NA |
| | | 2 | 1 | 1 | 3-4 | 2-4 | | **10** | **36** | **1786** | 29.79 | | 0 | 0 | NA |

## A.5 LOW-LEVEL RL POLICY

### A.5.1 RL TRAINING

We trained our model using PPO (Schulman et al., 2017) with learning rate $\alpha = 2.5 \times 10^{-4}$, clip parameter $\epsilon = 0.1$, value loss coefficient $c_1 = 0.5$ entropy coefficient $c_2 = 0.01$, GAE, recurrent policy, linear learning rate decay, 128 steps per update, 4 mini-batches. We train for 5 million steps on RTX 3080 GPU for 2 days. Observation Space (environment returns these at each step): RGB and Depth image. Agent position ($M^{2D}$ location + pitch + yaw). Object pick location for Pick policy training and object place location for training place policy. We also get

Table 4: Comparison on RoomR dataset

| Method | HOOP | Mirakhor et al. (2024a) [Table 2] | Mirakhor et al. (2024b) [Table 4] |
|---|---|---|---|
| Scene Success | 49 | 43 | 34 |

information if an action succeeded or failed from the AI2Thor environment. The training process for the pick model involves randomly positioning the target object within a specified proximity to the agent. The goal is to pick a selected object successfully. For the place model, the training method-ology follows a similar approach, with the key distinction being the absence of object detection requirements, as the agent begins each scenario already holding the object. In all training instances for the place model, the initial state consists of the agent holding an object, and the task involves depositing the object at a predetermined location.

### A.5.2   ACTION SPACE:

MoveAhead, MoveBack, MoveLeft, MoveRight, LookUp, LookDown, RotateLeft, RotateRight, PickObject (for Pick policy only), PlaceObject (for Place policy only)

We allow the Pick and Place policies to have navigation actions because we might not always be in the perfect position to interact with an object, and we want our policy to be able to handle these scenarios.

### A.5.3   REWARD FUNCTION:

Reward function: -1 for each navigation action, +50 for successful interaction action, -50 for failed interaction.

### A.5.4   TRANSITION FUNCTION:

The transition function in AI2-Thor is deterministic. For the navigation actions, a move action moves the agent in the selected direction by 1 unit. Rotation action rotates the agent by 45 degrees in the given direction. Look action titles the head of the agent by 30 degrees in the given direction.

## A.6   FRONTIER EXPLORATION + HAND CODED INTERACTION METHOD DETAILS

### A.6.1   OVERVIEW OF THE HEURISTIC APPROACH

Here, we provide a deeper dive into the construction of the hand-coded heuristic method. The heuristic method serves as an important baseline that replaces the sophisticated MCTS search planning of our HOO-POMDP with hand-coded expert strategies while retaining the same belief update apparatus. This approach helps us evaluate the specific contribution of principled planning in handling uncertainty during rearrangement tasks.

The algorithm presents our hand-coded heuristic method for object rearrangement tasks, serving as a critical baseline for comparison with the HOO-POMDP approach. As shown in Algorithm 4, this method maintains the same belief update apparatus while substituting sophisticated MCTS planning with an expert-designed strategy.

The agent alternates between two phases: exploration (lines 21-23) and object interaction (lines 12-19). During exploration, the agent identifies frontier clusters at the boundary between known and unknown space, navigating to the closest frontier to systematically discover objects. The object interaction phase is triggered when the belief probability for any object exceeds the confidence threshold $\theta$ (line 9, 70% for our experiments). When this occurs, the agent temporarily halts exploration to retrieve and place the object at its goal location. The navigation system uses A* and the PickPlace policy is the same as in our HOOP system (uses RL pick, A* navigation and RL place policy to achieve PickPlace).

---

**Algorithm 4:** Hand-Coded Heuristic Method for Object Rearrangement

---

**Input:** Environment $E$, Set of target objects $O$, Confidence threshold $\theta$ (default: 0.7)
**Output:** Rearranged objects at goal locations
Initialize belief state $B$ over object locations;
Initialize 2D occupancy grid map $M$ (initially all cells unknown);
Initialize empty set of frontier clusters $F$;
Initialize empty set of found objects Found;
**while** *not all objects in $O$ are rearranged* **do**
    Update belief state $B$ based on current observations;
    $F \leftarrow$ IdentifyFrontierClusters($M$);
    **foreach** *object $o \in O$ not yet rearranged* **do**
        **if** $\max(B(o)) > \theta$ **and** $o \notin$ *Found* **then**
            Add $o$ to Found;
    **if** *Found $\neq \emptyset$* **then**
        $o_{\text{closest}} \leftarrow$ GetClosestObjectFrom(Found);
        Remove $o_{\text{closest}}$ from Found;
        $\text{loc}_{\text{object}} \leftarrow \arg\max(B(o_{\text{closest}}))$;
        NavigateTo($\text{loc}_{\text{object}}$);
        PickPlace($o_{closest}$)
    **else**
        $f_{\text{closest}} \leftarrow$ GetClosestFrontierCluster($F$);
        NavigateTo($f_{\text{closest}}$);
        Update $M$ based on new observations;

---

This approach provides a direct evaluation of the contribution of principled planning in handling uncertainty during rearrangement tasks, as it isolates the planning component while maintaining identical belief representations.

Since this is a simple system,it is limited to basic rearrangement and cannot be generalized to more complex problems - such as blocked goals. We would have to hand-design a new policy for each new case whereas our HOOP system can handle and plan for new scenarios much more easily.

### A.6.2 CONFIDENCE THRESHOLD SELECTION

The 70% confidence threshold was carefully calibrated for the specific characteristics of our problem:

1. **Higher thresholds ($>$70%)**: With our imperfect object detector (50-60% success rate), setting a higher confidence threshold would result in many objects never exceeding the threshold even when directly observed multiple times. This would lead to excessive exploration and low task completion rates.

2. **Lower thresholds ($<$70%)**: Setting a lower threshold increases the risk of false positives, where the agent attempts to interact with an object that isn't actually present at the believed location. Our implementation treats failed pick attempts as definitive evidence that the object is not present, and the agent will not try again at that location. Therefore, false positives can permanently prevent successful rearrangement of certain objects.

3. **Empirical optimization**: The 70% threshold represents an empirically determined balance that maximizes overall task completion while minimizing both excessive exploration and false positive interactions.

### A.7 COMPONENT GLOSSARY FOR HOO-POMDP

This glossary provides standardized terminology for the key components of our Hierarchical Object-Oriented POMDP (HOO-POMDP) approach to help readers maintain a clear understanding throughout the paper.

### A.7.1 Core System Components

| Term | Definition | Algorithm 1 Reference |
|------|-----------|------------------------|
| **HOO-POMDP** | The complete hierarchical planning system for object rearrangement in partially observable environments. | Full Algorithm 1 |
| **Perception Subsystem** | Processes RGB and depth images to detect objects and generate observations. | `PERCEPTIONSYSTEM()` (line 8) |
| **Belief Update Subsystem** | Maintains and updates probability distributions over possible object locations.

Fully described in Algorithm 2 | `BELIEFUPDATE()` (line 9) |
| **Abstraction System** | Converts continuous low-level belief state into discrete abstract state. | `GENERATEABSSTATE()` (line 10) |
| **Abstract OOPOMDP Planner** | Uses POUCT to search through abstract actions and find the best sub-goal. | `POUCTPLANNER()` (line 11) |
| **Policy Executor** | Converts abstract sub-goals into a sequence of executable low-level actions.

It also executes the first low-level action in the above sequence in the environment. | `GETLOWLEVELPOLICY()` (line 14)
`lowLevelPolicy.GetAct` (Line 15)
`ENV.EXECUTE()` (line 7) |

### A.7.2 Key Data Structures

| Term | Definition | Representation |
|------|-----------|----------------|
| **Belief State** | Probability distribution over possible locations for each object. | $b = \prod_{i=1}^{n} b_i$ |
| **Abstract State** | Discrete representation of the world state used by the planner. | $s = (s_r, s_{targets})$ |
| **Observation** | Detection of object locations from the perception system. | $z = (z_{robot}, z_{objects})$ |
| **2D Occupancy Map** | Discretized grid representation of the navigable space. | $M^{2D}$ |

### A.7.3 Key Actions and Policies

| Term | Definition | Examples |
|------|-----------|----------|
| **Abstract Actions** | High-level actions output by the abstract planner. | MoveAB, Rotate$_{angle}$ |
| **Sub-goals** | Task-oriented goal states for low-level policies to achieve. | PickPlaceObject$_{i-goalloc}$ |
| **Low-level Actions** | Primitive actions executed directly in the environment. | MoveAhead, PickObject |
| **Low-level Policies** | Controllers that translate abstract actions into action sequences. | Move, Rotate, PickPlace |

### A.7.4 ALGORITHM COMPONENTS

| Term | Definition | Reference |
|------|-----------|-----------|
| **POUCT** | Partially Observable UCT - search algorithm for the abstract planner. | Algorithm 3 |
| **Frontier Exploration** | Strategy to systematically explore unknown areas. | Appendix A.6 |
| **A$^*$ Algorithm** | Path planning algorithm for navigation between locations. | Policy Executor (line 248) |

### A.7.5 OBJECT STATE REPRESENTATION

| Term | Definition | Components |
|------|-----------|-----------|
| **Object State** | Complete representation of an object in the abstract state. | $(loc_i, pick_i, placeloc_s,$ $is\_held, at\_goal, g_i)$ |
| **Pick Location** | Location from which an object can be picked up. | $pick_i$ |
| **Place Locations** | Set of locations where an object can be placed. | $placeloc_s$ |
| **Goal Location** | Target location for task completion. | $g_i$ |

### A.8 LLM USAGE

LLMs were used to correct grammar. They were also used as coding aids.

