# OpenReview forum: "Hierarchical Object-Oriented POMDP Planning for Object Rearrangement"
_ICLR.cc/2026/Conference — Submitted to ICLR 2026_

### Official Review · Reviewer_v1Hc · 2025-10-31

**Soundness:** 3
**Presentation:** 3
**Contribution:** 2
**Rating:** 4
**Confidence:** 5

**Summary:**

The paper proposes a novel method for multi-room, multi-object rearrangement in a partially observable setting. The rearrangement process operates in two phases. In the walkthrough phase, the agent explores the environment to build a map, record receptacle locations, and gather scene information; objects are then placed randomly in the environment. In the rearrangement phase, the agent uses the constructed map, the known set of object classes, and the specified goal locations to perform the rearrangement. The approach employs a hierarchical planning framework, with a high-level planner based on PO-UCT to generate abstract actions such as move, pick, and place, and a low-level controller that uses reinforcement learning for pick-and-place behaviors and A* for navigation. The method explicitly addresses challenges posed by blocked paths and blocked goal locations, which earlier approaches do not handle effectively. Experimental results demonstrate that the system successfully operates under partial observability and scales to scenarios with up to 20 objects. Additionally, the paper introduces MultiRoomR, a benchmark featuring more rooms, more objects, and a greater variety of rearrangement configurations.

**Strengths:**

1. The paper is well-written and clearly structured, making the motivation and methodology easy to follow.
2. The proposed method is novel, particularly in how it discovers partially observable objects. It also effectively handles blocked-path scenarios, which prior work does not address.
3. The approach benefits from using PO-UCT as an iterative planner, which reduces dependence on learned policies, thereby lowering the risk of catastrophic failures often seen in purely neural methods.
4. The experimental evaluation is extensive and thorough, including strong comparisons to prior approaches and informative ablation studies that highlight the importance of each system component.
5. The newly introduced MultiRoomR benchmark is a valuable contribution that provides a more challenging and realistic testbed for future research on multi-room, multi-object rearrangement.
6. The authors have released the code and provided detailed algorithmic descriptions to support reproducibility, and they include the MultiRoomR dataset along with the supplementary material to facilitate future research.

**Weaknesses:**

1. Limited novelty in methodology: While the system is thoughtfully designed, the novelty is incremental. The method primarily combines existing components (PO-UCT, PPO-based low-level skills, and A* navigation) into a pipeline, and the core contribution lies in their integration. The main new challenge addressed is the blocked-path scenarios, which, although important, limits the conceptual innovation relative to prior hierarchical rearrangement frameworks.
2. Scalability concerns due to PO-UCT: The reliance on PO-UCT for high-level planning means computation scales with the number of simulations and search depth, which is strongly influenced by the number of objects and environment complexity. As a result, planning time may grow significantly as task complexity increases, limiting scalability to larger or more cluttered scenes.
3. Limited generalizability to real-world deployment: Because the high-level decisions are generated through Monte-Carlo tree search rather than a learned policy, the system must perform online planning at each step. This limits generalization and makes real-world application challenging, as the approach does not amortize planning into a neural policy that could execute quickly without repeated simulation.
4. Problem formulation relies on extra prior knowledge: The problem setup assumes access to the set of object classes that need to be rearranged, which is not typically available in prior works and may not be realistic in practical settings. Furthermore, the method assumes that the user provides precise 3D goal locations for objects. In real-world applications, such explicit spatial goal specifications are uncommon; instead, agents are generally expected to use commonsense semantic priors to infer plausible goal locations (e.g., as in TIDEE (Sarch et al.) or Housekeep (Kant et al.)). Integrating semantic goal inference would make the approach more practical and aligned with real-world deployment expectations.
5. Simplifying independence assumption in belief modeling: The belief update assumes objects are independent, which does not hold in real household environments where objects commonly co-occur, occlude each other, or exhibit relational structure (e.g., items inside containers or grouped functionally). This assumption restricts the model’s ability to reason about relational or compositional uncertainty and may limit performance in more complex, cluttered scenes.

**Questions:**

1. To better understand the scalability of the proposed approach, could you clarify the minimum number of simulations and the required search depth needed to achieve the best performance for different numbers of rooms and objects? Additionally, how does compute time grow as these parameters increase? A quantitative analysis of simulation budget vs. performance vs. time would help assess how the method scales to larger and more complex environments.
2. In the appendix Table 3, the “Time (m)” column is not clearly defined. Does this value represent the average total clock time to complete an entire rearrangement episode (planning + execution), or only the planning time? Additionally, is this reported per episode or normalized per object?
3. Have you considered incorporating semantic goal inference or preference-based goal discovery, similar to TIDEE or Housekeep?
4. Can the framework be extended to reason about correlations (e.g., object occlusions, functional groupings)? Do you observe failure cases where this assumption becomes limiting?
5. Can you provide insights into the most common failure modes? For example, is performance more sensitive to detection errors, navigation failure, or belief update ambiguity?
6. Could you comment on the feasibility of deploying this framework on a real robot?

---

> ### Author Response · Authors · 2025-11-29
> **Response to v1Hc Part 1**
>
> >Scalability concerns due to PO-UCT:
>
> Yes, it could take longer if we let the search go on until it finds an optimal solution. But we restrict our MCTS rollout depth and MCTS simulations - so it takes the same amount of time for each planning step irrespective of number of objects. This might lead to sub-optimal actions in the beginning as we do not have enough information to come up with good plans fast, but as we gather more information and update the belief, the planner comes up with good actions to take.
>
> > Limited generalizability to real-world deployment: Because the high-level decisions are generated through Monte-Carlo tree search rather than a learned policy, the system must perform online planning at each step. This limits generalization and makes real-world application challenging, as the approach does not amortize planning into a neural policy that could execute quickly without repeated simulation.
>
> While using a neural policy could make things faster, planners are much more generalizable -> for any starting or goal state configuration, given enough time to search, planners will give us a solution. But, the neural policies are not guaranteed to give a solution. It might be useful to incorporate learning to guide the planner itself, and is potential future work.
>
> >Problem formulation relies on extra prior knowledge. Integrating semantic goal inference would make the approach more practical and aligned with real-world deployment expectations.
>
> Yes, we do assume we have access to more information than previous works from one perspective -> the object classes that need to be rearranged. This is why we also coded a hand-designed policy to show the usefulness of a planner even when all the objects to be rearranged are known beforehand. Integrating semantic goals with our work is orthogonal to our contributions and doable - if a semantic goal such as, place on kitchen counter is provided, all we need to do is sample any location on the kitchen counter and our system can be used in the exact same way. Another way of incorporating a semantic goal could be, the low-level agents could be trained for them - suppose we train the place policy to take in as input the semantic goal, then we can input that whenever we chose to move the mug. Our modular framework allows to incorporate semantic goals at multiple levels.
>
> > In the appendix Table 3, the “Time (m)” column is not clearly defined. Does this value represent the average total clock time to complete an entire rearrangement episode (planning + execution), or only the planning time? Additionally, is this reported per episode or normalized per object?
>
> This is time averaged per solved problem. The time recorded the planning + execution time.
>
> >Have you considered incorporating semantic goal inference or preference-based goal discovery, similar to TIDEE or Housekeep?
>
> Incorporating goal inference is orthogonal to our work and straightforward to fit in. As this is about goal inference, and is external to the planner, the goal location information can come from anywhere. Given a set of objects to clean up, we could simply query a general LLM to ask where the goal locations for these objects could be in this world and start our planning based on the output from the LLM.
>
> >Can the framework be extended to reason about correlations (e.g., object occlusions, functional groupings)? Do you observe failure cases where this assumption becomes limiting?
>
> The framework can handle co-relations as long as they are not between 2 target objects. The framework needs to be updated for handling occlusions or functional groupings - as it would change the way the belief is managed - one of the ways could be, we assume objects are independent as long as they are far away and then only objects close to each other affect each other’s belief, in this way, we can keep the belief update efficient while handling occlusions etc.
>
> > Can you provide insights into the most common failure modes? For example, is performance more sensitive to detection errors, navigation failure, or belief update ambiguity?
>
> Please refer to response to Reviewer 2 (bVLn) for detailed breakdown of failures.
>
> > Could you comment on the feasibility of deploying this framework on a real robot?
>
> The framework’s modularity help make it feasible to deploy in the real world. For that, we would need to replace the low-level skills/policies with those of the real robot. When the low-level policies succeed, our system would work the same way without any changes as long as the low-level policies can achieve their sub-goals. If the policies, fail, then our system will simply try to re-plan but might run into issues there if the updated state is not visible (if an object was dropped while trying to move - detecting the object drop and getting it’s new fallen state is not part of the planning process).
>
> Results for Scalability with different number of simulations etc. are part of response part 2.

---

> ### Author Response · Authors · 2025-11-29
> **Response to v1Hc Part 2**
>
> > To better understand the scalability of the proposed approach, could you clarify the minimum number of simulations and the required search depth needed to achieve the best performance for different numbers of rooms and objects? Additionally, how does compute time grow as these parameters increase? A quantitative analysis of simulation budget vs. performance vs. time would help assess how the method scales to larger and more complex environments.
>
> For all settings, we used a maximum planning depth of 12. We used 500 simulations for all settings. Anything less than this, the system came up with poor actions often and we had to increase the number of simulations. Changing the planning depth beyond 12 did not make much of a difference.
>
> Results of number of simulations v/s performance v/s time (Current  HOOP uses 500 simulations):
>
> | Dataset | Objs | #BG | #Sw | #BP | #Rm | #V | SS↑ | OS↑ | TA↓ | Time | SS↑ | OS↑ | TA↓ | Time | SS↑ | OS↑ | TA↓ | Time |
> |---------|------|-----|-----|-----|-----|-----|-----|-----|-----|------|-----|-----|-----|------|-----|-----|-----|------|
> | | | | | | | | | 100 | | | | 500 | | | | 1000 | | |
> | RoomR | 5 | 1 | 0 | 0 | 1 | 3-4 | 41 | 62 | 248 | **0.54** | 49 | 71 | 211 | **1.61** | 49 | 72 | 205 | **2.58** |
> | Proc | 5 | 1 | 0 | 0 | 2 | 2-3 | 38 | 58 | 401 | **1.15** | 46 | 68 | 352 | **3.42** | 46 | 69 | 343 | **5.47** |
> | MultiRoomR | 10 | 1 | 1 | 0 | 2 | 2-3 | 24 | 54 | 798 | **2.63** | 32 | 65 | 710 | **7.89** | 32 | 66 | 691 | **12.62** |
> | | 10 | 2 | 1 | 1 | 2 | 2-3 | 14 | 39 | 876 | **3.01** | 21 | 49 | 789 | **8.98** | 22 | 51 | 768 | **14.37** |
> | | 10 | 2 | 1 | 0 | 3-4 | 1-2 | 21 | 51 | 1312 | **4.52** | 30 | 62 | 1189 | **13.45** | 29 | 60 | 1157 | **21.52** |
> | | 10 | 2 | 1 | 1 | 3-4 | 1-2 | 11 | 33 | 1458 | **5.34** | 18 | 44 | 1321 | **15.97** | 18 | 44 | 1285 | **26.35** |
> | | 15 | 1 | 1 | 0 | 3-4 | 2-3 | 14 | 47 | 1369 | **6.65** | 22 | 59 | 1228 | **19.89** | 21 | 57 | 1195 | **32.82** |
> | | 15 | 2 | 1 | 1 | 3-4 | 2-3 | 7 | 30 | 1564 | **7.41** | 14 | 41 | 1416 | **22.12** | 13 |  40 | 1378 | **36.50** |
> | | 20 | 2 | 1 | 0 | 3-4 | 2-4 | 9 | 42 | 1798 | **9.24** | 17 | 55 | 1621 | **27.61** | 17 | 55 | 1578 | **45.56** |
> | | 20 | 2 | 1 | 1 | 3-4 | 2-4 | 4 | 24 | 1965 | **9.98** | 10 | 36 | 1786 | **29.79** | 10 | 36 | 1738 | **49.15** |
>
> Analysis of MCTS Simulation Budget. We evaluated our approach across three simulation budgets (100, 500, 1000) to understand the trade-off between computational cost and performance. With 100 simulations, the planner frequently selects suboptimal actions, leading to substantially lower success rates across all metrics. While the lower TA values might appear favorable, this is misleading—the planner predominantly solves only simpler scenes while failing on more challenging configurations, often hitting the maximum step limit before task completion. The 500-simulation setting represents our primary configuration, achieving strong performance across all difficulty levels. Increasing to 1000 simulations yields marginal improvements: scene success remains unchanged, object success improves by only 1–2%, and task actions decrease by approximately 2–3%. However, computation time scales roughly linearly with simulation count, requiring 1.5–1.7× longer than the 500-simulation baseline. Given the minimal performance gains relative to the substantial increase in computational overhead, 500 simulations provides the optimal balance between efficiency and effectiveness for our task domains

---

### Official Review · Reviewer_mkKK · 2025-10-31

**Soundness:** 3
**Presentation:** 4
**Contribution:** 3
**Rating:** 6
**Confidence:** 3

**Summary:**

This paper proposes a hierarchical planning algorithm for the object-rearrangement problem, in the presence of partial observability. The three primary components of the algorithm are: 1) high-level POMDP planning in an abstract state space, 2) macro-actions which are implemented using Markov methods, and 3) an object-factored belief state. The result is a practical and high-performing algorithm for this important robot subtask. The algorithm is thoroughly evaluated on a novel benchmark task, where its performance exceeds previous methods'.

**Strengths:**

- This is an important problem that displays many of the characteristics of real-robot tasks.
- The approach is practical and simple, and builds on a lot of recent work focused on exploiting structure in POMDPs to generate efficient solutions.
- The method is sufficiently structured that I expect it would work on a real robot, with some engineering effort required obviously.
 - Using Markov low-level controllers to support a more abstract level that can then plan taking partial-observability into account has appeared in a few places, and has worked very well. This is a highly appropriate use of that strategy.
- The evaluation is thorough and the new benchmark proposed is valuable.
 - The paper is well written (though not well-structured, see below) and easy to read.

**Weaknesses:**

- The paper is oddly structured. There's a Related Work section that occurs before the Background section, which makes no sense. How am I to understand the RW section when I haven't been precisely told what problem you are solving yet? Then the Background section is called Problem Formulation, though instead of a formulation (which is a precise mathematical description of the task), we are given some implementation details about AI2Thor. The paper and ideas have to stand on their own outside of AI2Thor. There's some problem formulation going on in section 4 (!!), where the authors define an OO-POMDP. Anyway I had to re-read the paper several times, out of order, to actually follow, which is bad.
- The authors talk about embodied AI, but there are no bodies in this paper. Also, the first sentence is a little melodramatic. I'm not sure that multi-object rearrangement in a simulator is such a fundamental challenge. It's a good problem to work on though. Anyway I would tone some of that down.
 - It's not clear what the authors mean by "hand-coded" planning approach. Do they mean someone hand-codes a plan? Or a planner? Hand-coding a planner is what the planning research community does all day! Or do they mean, hand-design for a specific case? In that case, that's what this paper does! But they seem to be criticizing prior work here, so I think some clarify about what exactly they are criticizing would be good.
 - The paper is related to structured models of POMDPs. It cites an extends OO-POMDPs, which is appropriate, but I think MOMDPs are Merlin's work on local observability is probably also relevant.
 - Occasionally the paper uses a parenthetical citation as a noun, which should be fixed.

**Questions:**

In the table, why does PK not have a 100% success rate? Or maybe that's not a percentage?

Can you please confirm my understanding that the low-level policies for the macro-actions are essentially Markov? Are there any implications for that>

---

> ### Author Response · Authors · 2025-11-29
> **Response to reviewer mkKK**
>
> >The paper is oddly structured. There's a Related Work section that occurs before the Background section, which makes no sense. How am I to understand the RW section when I haven't been precisely told what problem you are solving yet? Then the Background section is called Problem Formulation, though instead of a formulation (which is a precise mathematical description of the task), we are given some implementation details about AI2Thor. The paper and ideas have to stand on their own outside of AI2Thor. There's some problem formulation going on in section 4 (!!), where the authors define an OO-POMDP. Anyway I had to re-read the paper several times, out of order, to actually follow, which is bad.
>
> Thank you for the pointing it out. We will add more details about the problem being solved in the introduction to better relate to related work. We will add background from section 4 before problem formulation, and add a clear problem formulation statement in that section.
>
> > The authors talk about embodied AI, but there are no bodies in this paper. Also, the first sentence is a little melodramatic. I'm not sure that multi-object rearrangement in a simulator is such a fundamental challenge. It's a good problem to work on though. Anyway I would tone some of that down.
>
> Thank you for pointing it out. We will tone it down.
>
> >It's not clear what the authors mean by "hand-coded" planning approach. Do they mean someone hand-codes a plan? Or a planner? Hand-coding a planner is what the planning research community does all day! Or do they mean, hand-design for a specific case? In that case, that's what this paper does! But they seem to be criticizing prior work here, so I think some clarify about what exactly they are criticizing would be good.
>
> By hand-coded we mean, a sequence of high-level actions is encoded by a domain expert. For example, In the paper "Multi-skill Mobile Manipulation for Object Rearrangement", a plan is hand-coded beforehand - move to object, pick up object, move to goal location, place object. Only which object is being interacted is decided based on what is closest and will be moved first. The planner is only deciding what object, but the sequence In which the low-level policies are called are fixed beforehand
>
>
> >The paper is related to structured models of POMDPs. It cites an extends OO-POMDPs, which is appropriate, but I think MOMDPs are Merlin's work on local observability is probably also relevant.
>
> We can add that for better coverage of related work. Thank you.
>
> >Occasionally the paper uses a parenthetical citation as a noun, which should be fixed.
>
> Thank you for pointing it out. We will fix it.
>
> >In the table, why does PK not have a 100% success rate? Or maybe that's not a percentage?
>
> Ans :It is a percentage.  PK is not 100% because the low-level interactions - such as pick/place are not 100% successful. So, even when we know exactly where an object is, we can fail to pick it up sometimes. We use a RL based agent for both pick and place. If an object is far away (on a table and hence we cannot move closer), it needs to be accessed from a certain location only - the pick will fail otherwise. Similarly for places, it is not always perfect - if we try to place an object from a slightly off position from the perfect position, then it can be placed somewhere else by the simulator and is considered a failure.
>
> > Can you please confirm my understanding that the low-level policies for the macro-actions are essentially Markov? Are there any implications for that.
>
> Yes, low level policies are Markov. It means, we can use A* for navigation as we know the exact outcome of move actions and do not need to look back more than current state to decide the next action and same with the interaction actions - our current state tells us everything on what needs to be done to achieve the sub-goal. The main implication is that we can use simple independent systems for each of the low-level policies - A* and RL, and not have to worry about history while planning, simplifying the job of the low-level policies.

---

### Official Review · Reviewer_bVLn · 2025-11-07

**Soundness:** 2
**Presentation:** 1
**Contribution:** 1
**Rating:** 2
**Confidence:** 2

**Summary:**

The authors propose a hierarchical solution for an Object-Oriented Partially Observable Markov Decision Process (OOPOMDP). The system decomposes the problem by using a high-level abstract POMDP planner (based on PO-UCT) to reason over object-factored belief states and generate high-level actions (e.g., MoveToLocation, PickPlaceObject). A low-level policy executor, composed of classical planners and trained RL policies, is then responsible for executing these sub-goals. They also built a multi-room evaluation on top of ProcThor, creating a new multi-room dataset for evaluation.

**Strengths:**

The authors demonstrate an improvement over their selected baselines of FHC, VRR, and MSS. I found the inclusion of different ablations, such as having an oracle belief or a perfect object detector, helps validate their design choices. Their multi-room benchmark also highlights the limitations of the methods they chose to compare against.

**Weaknesses:**

My main concerns are as follows:
* The object independence assumption seems to directly conflict with trying to improve performance on scenarios involving blocked goals or blocked paths.
* Requiring a pre-task walkthrough to build a map of the static environment is a significant practical limitation. This makes the solution inapplicable to new environments and the methodologies' robustness to any change in the map, or if there were blocking objects during this phase of planning.
* The error analysis clearly identifies that the low-level pick/place policies are a major bottleneck, which makes it hard to isolate the planning performance.
* The manual initialization of possible abstract actions (described in Abstract OOPOMDP Planner) limits the system's generality. The planner does not discover actions but instead searches over a pre-defined set based on object locations. This would limit the tasks to which this can be applied.
* The paper's presentation is often unclear, making it difficult to fully grasp the methodology and its positioning within related works. Key details of the planning and abstraction systems are distributed between the main text and the appendix, hindering readability.
* The anonymous code link provided in the paper is expired.

**Questions:**

* How does FHC perform with an Oracle object detector?
* Do other methods require an initial walk-through of the environment?
* How sensitive is the planner to an imperfect static map from the "walkthrough phase"? What happens if a static object, like a chair, is moved between the walkthrough and the rearrangement phase?
* Given that low-level policy failures are a key bottleneck, do you have any data on the planner's success rate? For example, during failed episodes, does the planner still produce a semantically correct sequence of abstract sub-goals?
* Could you clarify the algorithmic novelty of your algorithm's planner? The paper positions it as an extension of OO-POMDPs from search to rearrangement. Is the primary difference simply the inclusion of PickPlace actions and their corresponding belief updates?

---

> ### Author Response · Authors · 2025-11-29
> **Response to Reviewer bVLn**
>
> >The object independence assumption seems to directly conflict with trying to improve performance on scenarios involving blocked goals or blocked paths.
>
> Object independence assumption is about observations and beliefs about the object positions. So, when we see an object, we assume that the visibility of this object is not affected by any other object. The belief is only over object location and not its accessibility and for belief maintenance and update purposes, the objects are independent.
> Object dependence comes in when talking about accessibility. When we move one object, another object might becomes accessible and we do not assume object independence here - and it does not affect our belief update or our search.
>
> >Requiring a pre-task walkthrough to build a map of the static environment is a significant practical limitation. .
>
> For any new environment, the walkthrough needs to be done once to get information about non-movable objects. Common example is when a robot is deployed in a household - it needs to map the house just the first time. From there on, whenever it needs to clean the house it can use this map.
>
> >Error Analysis. During failed episodes, does the planner still produce a semantically correct sequence of abstract sub-goals?
>
> Yes, even during failed episodes, on most occasions, the planner does produce semantically correct sequence of sub-goals.
>
> Error analysis :
>
> | Dataset | Objs | BP | #Rm | Total | Detector | Controller | Planning |
> |---------|------|----|-----|-------|-----|------|-------|
> | RoomR | 5 | 0 | 1 | 29 | 8 | 20 | 1 |
> | | | 1 | | 39 | 9 | 26 | 4 |
> | Proc | 5 | 0 | 2 | 32 | 9 | 19 | 4 |
> | | | 1 | | 47 | 10 | 31 | 6 |
> | MultiRoomR | 10 | 0 | 2 | 35 | 7 | 25 | 3 |
> | | | 1 | | 51 | 9 | 39 | 3 |
> | | 10 | 0 | 3-4 | 38 | 7 | 28 | 3 |
> | | | 1 | | 56 | 13 | 39 | 4 |
> | | 15 | 0 | 3-4 | 41 | 8 | 30 | 3 |
> | | | 1 | | 59 | 12 | 44 | 5 |
> | | 20 | 0 | 3-4 | 45 | 11 | 32 | 4 |
> | | | 1 | | 64 | 15 | 46 | 5 |
>
> > The manual initialization of possible abstract actions. This would limit the tasks to which this can be applied.
>
> We do initialize actions - we are only defining schemas. Action initialization depends on the problem we are solving and current state. Also, planning generally requires us to know the model of the world and the planner uses this model to search, discovering actions is a very different area of research and out of scope of our work. We are trying to make an agent better at rearrangement and our hierarchical definition and abstraction system help make that happen.
>
> > Paper presentation clarity:
>
> We will update the following to make it clearer: We will add more details about the problem being solved in the introduction to better relate to related work. We will add background from section 4 before problem formulation, and add a clear problem formulation statement in that section.
>
> >Link Expiry .
>
> We have fixed the link. It is accessible now.
>
> >How does FHC perform with an Oracle object detector?
>
> Its success rate is similar to PD but takes more steps.
>
> | Dataset | Objs | #BP | #Rm | #V | SS↑ | OS↑ | TA↓ | SS↑ | OS↑ | TA↓ |
> |---------|------|-----|-----|-----|-----|-----|-----|-----|-----|-----|
> | | | | | | FHC | | | FHC-Oracle | | |
> | RoomR | 5 | 0 | 1 | 3-4 | 38 | 58 | 269 | 59 | 83 | 207 |
> | Proc | 5 | 0 | 2 | 2-3 | 32 | 61 | 411 | 59 | 79 | 311 |
> | MultiRoomR | 10 | 0 | 2 | 2-3 | 20 | 44 | 931 | 38 | 75 | 670 |
> | | 10 | 1 | 2 | 2-3 | 12 | 38 | 993 | 28 | 65 | 698 |
> | | 10 | 0 | 3-4 | 1-2 | 19 | 34 | 1345 | 37 | 74 | 921 |
> | | 10 | 1 | 3-4 | 1-2 | 9 | 26 | 1490 | 29 | 67 | 1011 |
> | | 15 | 0 | 3-4 | 2-3 | 12 | 31 | 1605 | 29 | 71 | 1056 |
> | | 15 | 1 | 3-4 | 2-3 | 7 | 23 | 1886 | 25 | 69 | 1098 |
> | | 20 | 0 | 3-4 | 2-4 | 0 | 18 | NA | 20 | 18 | 1369 |
> | | 20 | 1 | 3-4 | 2-4 | 0 | 11 | NA | 16 | 10 | 1489 |
>
> >Do other methods require an initial walk-through?
>
> Yes, all methods require an initial walk-through of the environment
>
> >How sensitive is the planner to an imperfect static map from the "walkthrough phase"?
>
> It can handle when a static object is moved. When a static object is moved, it will be updated in the map when it comes in view -> if this movement makes a part of the room inaccessible, then our planner will realize that and end after moving all the accessible objects. If it does not, then we simply avoid this object like all other static objects and move around.
>
> >Could you clarify the algorithmic novelty of your algorithm's planner?
>
> Yes, inclusion of inclusion of PickPlace actions and their corresponding belief updates, is our main contribution from the planner perspective. The other contributions of the paper are :
> 1. The belief state abstraction system that converts the low-level belief representation into an abstract state that can be used by the planner.
> 2. MultiRoomR dataset - which provides a diverse and challenging set of scenarios with a higher number of objects and rooms to test agent’s ability to handle scale and partial observability.

---

### Official Review · Reviewer_ZgC5 · 2025-11-10

**Soundness:** 2
**Presentation:** 3
**Contribution:** 2
**Rating:** 2
**Confidence:** 5

**Summary:**

This paper presents a Hierarchical Object-Oriented Partially Observable Markov Decision Process (HOO-POMDP) planner for multi-object (scene) rearrangement problems. This includes a Object-Oriented POMDP planner for generating sub-goals, with low-level policies for achieving these sub-goals and a method for converting the low-level world into a representation for abstract planning. In addition, the paper presents a benchmark simulation environment (called MultiRoomR) of multi-room environments with different levels of partial observability, obstructions, and multiple objects. The authors present results of experimentally evaluating the HOO-POMDP against baselines in the context of various benchmarks.

**Strengths:**

(i) The paper focuses on the problem of multi-object rearrangement, and formulates it as a probabilistic sequential decision-making problem under partial observability.

(ii) The paper provides a new benchmark for multi-object rearrangement in the form of multi-room environments.

**Weaknesses:**

(i) One key problem with the paper is that it makes claims that are not fully substantiated. For example, the authors mention that object rearrangement solutions are based on RL and hand-coded planning methods; it is not clear what the authors mean by "hand-coded planning methods", but there are many other ways of solving this problem. Also, existing probabilistic planners can handle large domains and uncertainty resulting from partial observability.

(ii) The discussion of related work unfortunately seems to be limited to the more recent RL-based or deep networks-based methods; there is no acknowledgement or discussion of the rich literature in hierarchical planning, including those based on RL and POMDPs, e.g., [1-3]. The existing papers have already explored  factorization of state space (and belief updates) in complex domains. The authors need to clarify how their proposed approach is different and makes a new contribution.

(iii) Following up on previous points: it is unclear if/how the proposed approach is an hierarchical planner. The proposed pipeline processes sensory inputs to obtain concepts that are then used to compute a plan for the rearrangement task. This is the standard pipeline in many AI systems that process sensor inputs and plan actions. The "abstraction system" is just a manually encoded method that is executed to map the sensor inputs to the representation for POMDP planning. This can be contrasted with prior work on hierarchical planning with POMDPs [1, 2] where there is an actual hierarchy of POMDPs and tasks to be performed. In addition, these systems support complex state spaces, uncertainty in perception and actuation, different objective functions (e.g., optimize for travel distance and time), and have been used for planning on physical robot platforms.

(iv) From the description of problem formulation and the algorithms, it is not clear if there is any actual uncertainty in actuation; even the uncertainty in perception is captured by a very simple model, and the "partial observability" seems to be a reference to the fact that the agent's view of the domain is limited at any point in time. It is also not clear why RL policies are needed for the low-level execution of simple movements in a simulation environment.

(v) In the experimental evaluation, the baselines seem to be systems that pose the rearrangement task as a learning problem (e.g., with deep networks), or seek to complete the task after removing the limited uncertainty introduced in perception. It is also not clear what it means to "remove the hierarchical planning" in the HOOP-HP baseline, and how it impacts the corresponding state (also action, observation) space. Given the limited noise in the system, it is unclear why there is no comparison with a state of the art classical planning or probabilistic planning system. Finally, the statistical significance of the results shown in Table 1 is unclear.

[1] Joelle Pineau and Sebastian Thrun. High-level Robot Behavior Control using POMDPs. National Conference on Artificial Intelligence (AAAI), 2002.

[2] Mohan Sridharan, Jeremy Wyatt and Richard Dearden. Planning to See: A Hierarchical Approach to Planning Visual Actions on a Robot using POMDPs. Artificial Intelligence, 174 (11):704-725, 2010.

[3] Harsha Kokel, Sriraam Natarajan, Balaraman Ravindran, and Prasad Tadepalli. RePReL: A Unified Framework for Integrating Relational Planning and Reinforcement Learning for Effective Abstraction in Discrete and Continuous Domains. Neural Computing and Applications, 35: 16877-16892, 2023.

**Questions:**

Please address comments in the "weaknesses" section above.

---

> ### Author Response · Authors · 2025-11-29
> **Response to Reviewer ZgC5 Part 1**
>
> >Clarification of "Hand-Coded Planning Methods"
>
> By "hand-coded planning," we refer to systems where a domain expert pre-specifies the sequence of high-level actions. For example, in "Multi-skill Mobile Manipulation for Object Rearrangement," the plan structure is fixed beforehand: move to object → pick up object → move to goal → place object. The system only decides which object to interact with (typically the closest), but the sequence in which low-level policies are invoked is predetermined. Such approaches cannot handle blocked paths or goal interference, where the optimal action sequence depends on the current configuration—moving Object B before Object A may be necessary even if A is closer.
>
>
> > Related Work Distinction
>
> > Distinction from Pineau & Thrun [1]
>
> Their approach partitions actions into a fixed hierarchy at design time, with abstract action parameters computed offline using exact POMDP solvers. This cannot apply to rearrangement for several reasons. First, actions like `PickPlace(Object_A, Location_X)` require dynamic instantiation—they cannot be parameterized without knowing Object A's location, which is discovered at runtime, and Location X's accessibility, which depends on other objects. Second, their approach assumes subtasks are sufficiently small to be solved exactly, but our O(L^n) state space for n objects admits no such decomposition. Third, their domain uses identity transition functions, whereas our PickPlace actions are world-altering and create cascading dependencies such as blocked paths, requiring online replanning.
>
> > Distinction from Sridharan et al. [2]
>
> Their visual planning exploits translation invariance to learn convolutional policy kernels for single-target search. This is inapplicable to our problem for several reasons. Manipulating Object A depends on the global configuration, specifically which objects block paths and goals, meaning there is no spatial invariance to exploit. Additionally, their observations do not change state, whereas our manipulations fundamentally alter configurations. Finally, a greedy entropy-reduction objective cannot handle sequential dependencies—clearing a blocked path has low immediate reward but enables future actions.
>
> > Distinction from Kokel et al. [3] (RePReL)
>
> RePReL uses symbolic planning to decompose goals into options, then learns RL policies with D-FOCI-based abstraction. The key differences are as follows. RePReL's planner assumes full observability, whereas we reason about belief states when object locations are unknown. Furthermore, D-FOCI captures single-object influences, but rearrangement requires joint reasoning where Object A's manipulability depends on Object B's position.
>
> Our contribution uniquely combines object-factored belief updates, dynamic action instantiation, online planning over manipulation actions, and explicit handling of inter-object dependencies.
>
> ---
>
> >  Clarification of Hierarchical Planning Structure
>
> > Our Hierarchy: Action Abstraction, Not Nested POMDPs
>
> Our hierarchy differs from [1,2] but follows the established options framework tradition. The abstract POMDP planner reasons over sub-goals such as Move, Rotate, and PickPlace without modeling low-level dynamics. Low-level policies execute these sub-goals using methods like A* and RL without knowledge of the high-level task structure. Information flows bidirectionally: belief updates propagate upward while sub-goals propagate downward. This constitutes hierarchical planning through temporal abstraction.
>
> > The Abstraction System is Computationally Non-Trivial
>
> Unlike static sensor-to-representation mapping, our abstraction performs runtime computation. Dynamic action instantiation generates valid PickPlace actions only when objects are detected with sufficient confidence. Accessibility reasoning checks reachability of pick and place locations, discarding blocked options based on the locations of objects in 3D space. This belief-dependent abstraction is what enables our system to handle inter-object dependencies that static mappings cannot address.
>
> We handle greater state complexity and observation uncertainty. We acknowledge physical deployment as future work and will clarify our hierarchical framing in the revision.

---

> ### Author Response · Authors · 2025-11-29
>
> > Uncertainty Sources and RL Policy Necessity
>
> Sources of Uncertainty
>
> There is no uncertainty in actuation. The partial observability arises from two distinct sources. First, the agent only observes a portion of the environment at any time due to its limited field of view. Second, object detectors fail to detect objects even when they are within the agent's view. This is a realistic challenge—perfect detectors do not exist in practice. Our per-class detection rates range from TP=0.015 (ladle) to TP=0.745 (plunger), reflecting real detector behavior.
> The simplicity of our observation model is intentional—it captures real-world detector characteristics (true positive/false positive rates, distance-dependent detection) while remaining tractable for belief updates. This design choice makes our approach more readily transferable to physical systems.
>
> >Necessity of RL for Low-Level Execution
>
> RL policies are required because AI2Thor's pick and place operations are not trivially executable. Objects must be accessed from specific positions and orientations—for example, an object on a table cannot be picked from arbitrary locations. The agent must position itself correctly, and picking will fail otherwise. Even with correct positioning, pick/place actions can fail due to occlusions, object geometry, or reaching constraints. The RL agent learns to handle these failure modes through retry strategies and position adjustments. Additionally, the RL agent outputs sequences of look, move, and interact actions to successfully complete manipulation—this is non-trivial even in simulation.
>
> >HOOP-HP Baseline Explanation
>
> Removing hierarchy means the POMDP planner operates directly over low-level actions (MoveAhead, RotateLeft, PickObject, etc.) rather than abstract actions (Move_AB,etc.). The state and belief spaces remain identical—only the action space changes. This dramatically increases the planning horizon, as each abstract action corresponds to tens or hundreds of low-level actions. The poor performance of HOOP-HP shows that low-level action planning is intractable at our problem scale.
>
> >Perception Uncertainty is Not Limited
>
> We respectfully disagree that perception uncertainty is limited. Detector failure rates are substantial and class-dependent (TP: 0.015–0.745). Repeated detection failures decrease belief about object presence, potentially causing the planner to abandon areas permanently. False positives cause manipulation of incorrect objects. These compound failures create significant challenges that baselines cannot handle—as evidenced by their poor performance.
>
> > Classical and Probabilistic Planning Comparison
>
> Classical planners assume full observability and cannot handle uncertain object locations from detector failures. Regarding probabilistic planners, POUCT is itself a state-of-the-art probabilistic planning algorithm. We did not compare against particle-based methods like POMCP because they lack factored belief representations—without object-factored updates, POMCP does not scale beyond 3-5 objects in a single room, making comparison on MultiRoomR (10-20 objects, 2-4 rooms) infeasible.
>
> > Statistical Significance
>
> We did 5 trial runs of our system (100 scenes each time) along with the ablation and oracle setting. We did not run VRR, FHC or MSS as they do not have randomness in their runs. We provide the mean and 95% confidence interval of the runs below.
>
> | Dataset | Objs | #BP | #Rm | #V | HOOP SS↑ | HOOP OS↑ | HOOP TA↓ | HOOP-HP SS↑ | HOOP-HP OS↑ | HOOP-HP TA↓ | PD SS↑ | PD OS↑ | PD TA↓ |
> |---------|------|-----|-----|-----|----------|----------|----------|-------------|-------------|-------------|--------|--------|--------|
> | RoomR | 50 | 1 | 3 | - | 448.8±1.04 | 71.4±2.26 | 215±12.90 | 13.0±2.64 | 33.2±5.02 | 320±56.90 | 62.4±1.25 | 87.4±1.87 | 195±7.80 |
> | Proc | 50 | 2 | 2 | - | 346.0±1.96 | 69.0±2.78 | 360±21.60 | 8.8±2.30 | 29.2±6.24 | 420±78.50 | 60.4±1.21 | 83.0±1.70 | 276±11.04 |
> | MRoomR | 100 | 2 | 2 | - | 332.2±1.04 | 65.4±2.26 | 720±43.20 | 4.8±2.36 | 25.2±6.42 | 1060±99.66 | 40.0±0.80 | 79.0±1.58 | 548±21.92 |
> |  | 101 | 2 | 2 | - | 321.2±1.04 | 49.4±2.26 | 815±48.90 | 2.2±1.42 | 19.2±6.36 | 1140±101.44 | 28.4±0.57 | 69.0±1.38 | 603±24.12 |
> |  | 100 | 3-4 | 1 | - | 230.2±1.04 | 62.4±2.26 | 1200±72.00 | 3.0±1.70 | 16.2±6.28 | 1400±137.84 | 37.0±0.74 | 75.0±1.50 | 859±34.36 |
> |  | 101 | 3-4 | 1 | - | 218.2±1.04 | 44.4±2.26 | 1355±81.30 | 1.2±0.54 | 7.0±4.62 | 1560±116.34 | 30.6±0.61 | 70.0±1.40 | 1014±40.56 |
> |  | 150 | 3-4 | 2 | - | 322.4±2.99 | 60.4±5.59 | 1250±75.00 | 0±0 | 4.8±2.48 | NA | 30.4±0.61 | 75.4±1.51 | 949±37.96 |
> |  | 151 | 3-4 | 2 | - | 314.2±2.96 | 43.0±4.73 | 1450±87.00 | 0±0 | 5.8±2.74 | NA | 25.0±0.50 | 71.0±1.42 | 999±39.96 |
> |  | 200 | 3-4 | 2 | - | 416.4±2.57 | 57.6±5.73 | 1650±99.00 | 0±0 | 4.8±2.30 | NA | 27.0±0.54 | 75.4±1.51 | 1238±49.52 |
> |  | 201 | 3-4 | 2 | - | 410.0±2.78 | 38.4±4.17 | 1800±108.00 | 0±0 | 3.8±2.56 | NA | 20.0±0.40 | 69.6±1.39 | 1378±55.12 |

---

### Meta-Review · Area_Chair_7EAQ · 2025-12-15

**Summary:**

The paper proposes a hierarchical object oriented POMDP formulation for multi-object rearrangement problems in partially observable multi-room environments. A new benchmark dataset is also presented. Reviewers did not find the method sufficiently novel and compelling.

**Reviewer Concerns:**

One reviewer is generally positive. Other reviewers were concerned about novelty and contribution of the paper. The authors provided detailed rebuttal but I do not think the views on novelty would change sufficiently.

**Reviewer Scores:**

The authors provided detailed rebuttal, which may have changed some scores. However, overall, I do not think the responses would have sufficiently changed the reviewers' initial views.

---

### Decision · Program_Chairs · 2026-01-26

Reject